# NTK with Convex Two-Layer ReLU Networks

## ABSTRACT

We theoretically analyze a convex variant of two-layer ReLU neural networks and how it relates to the standard formulation. We show that the formulations are equivalent with respect to their output values for a fixed dataset and also behave similarly during gradient-based optimization as long as the weights on the first layer of standard networks do not change too much, which is a common assumption for their convergence to an arbitrarily good solution. We further show that for any two-layer ReLU neural network, even considering those of infinite width, there exists a (weighted) network of width $O(n^{d-1})$ with the same output value on all data points. Furthermore, these finite networks have exactly the same eigenvalues $\lambda$ of their neural tangent kernel (NTK) matrix and the same NTK separation margin $\gamma$ as in the infinite width limit. After handling these preliminaries, we get to our main results: We give a $(1 \pm \epsilon)$ approximation algorithm for the separation margin $\gamma$ which was not known how to evaluate in general and we study two data examples: 1) a circular example for which we strengthen an $\Omega(\gamma^{-2})$ lower bound against previous worst-case width analyses; 2) a hypercube example that can be perfectly classified by the convex network formulation but not by any standard network, distinguishing their expressibility.

## 1 INTRODUCTION

The theory of neural networks is an active field of research with intriguing open questions. One important direction studies relatively 'simple' neural networks with only two layers, which allows a tractable formal treatment, while so called *universal approximator* theorems (Shalev-Shwartz and Ben-David, 2014) ensure that their expressive power is not limited compared to deep networks when two-layer networks are sufficiently wide. A property of two-layer neural networks with ReLU activation is that in the infinite width limit their kernel matrix and corresponding feature mapping computed on the first layer become stationary and the optimization problem becomes convex, given that the loss function used for training is convex (Jacot et al., 2018). This holds for an array of popular loss functions such as squared loss for regression and logistic loss for classification. A standard gradient descent can thus optimize up to arbitrarily small error in neural tangent kernel (NTK) space (Jacot et al., 2018). Unfortunately, convexity does not persist when their width is finite. The main issue lies in changing neuron activations when the network parameters are adjusted during optimization. However, gradient descent (GD) is popular and surprisingly successful for non-convex neural network optimization. It is thus very important to find theoretical explanations for this phenomenon (Li and Liang, 2018). To this end, our paper analyzes theoretically a formulation that we call *convex two-layer ReLU neural networks*. They are almost equivalent to the standard non-convex formulation under mild assumptions to be specified later. In particular, they require the same size and width as the standard formulation up to a factor of two. Thus any bound on their width implies a bound for the standard setting as well. At the same time they allow significantly simplified theoretical analyses since by decoupling the weight vectors to be trained from the orientation vectors that determine neuron activations, the training problem is convex for arbitrary convex loss functions.

**Related work** An ever-growing series of theoretical works study necessary and sufficient conditions on the width of standard two-layer ReLU networks such that GD converges to arbitrarily small error despite the non-convex optimization problem.

A large body of work studied convergence results for over-parameterized neural networks (Li and Liang, 2018; Du et al., 2019c; Allen-Zhu et al., 2019c;b; Du et al., 2019b; Allen-Zhu et al., 2019a;

Song and Yang, 2019; Arora et al., 2019b;a; Cao and Gu, 2019b; Zou and Gu, 2019; Du et al., 2019a; Lee et al., 2020; Huang and Yau, 2020; Chen and Xu, 2020; Brand et al., 2021; Li et al., 2021; Song et al., 2021). Early works proved the first finite upper bounds on the network width $m$, i.e., the number of neurons in the hidden layer (Li and Liang, 2018). (Du et al., 2019a) achieved $m = O(\lambda^{-4}n^6)$, where $\lambda$ denotes the smallest eigenvalue of the (NTK) kernel matrix, and $n$ is the number of input points. This was improved to $m = O(\lambda^{-4}n^4)$ (Oymak and Soltanolkotabi, 2020; Song and Yang, 2019).

Under the *additional assumption* that the data points are pairwise almost orthogonal, (Song and Yang, 2019) improved to $m = O(\lambda^{-4}n^2)$. Various data distributions such as uniform points on the sphere, or random hypercube vertices yield similar properties. Remarkably, if the standard Gaussian initialization of the weights on the first layer is *coupled*, where every Gaussian vector is copied once with positive and once with negative sign, then even $m = O(n/d)$ could be shown under the aforementioned data distributions (Oymak and Soltanolkotabi, 2020; Fiat et al., 2019; Daniely, 2020), or similar assumptions (Bubeck et al., 2020). (Kawaguchi and Huang, 2019; Zhang et al., 2021) also claim linear bounds in restricted settings, but no general guarantees.

It is important to continue work on the width of two-layer neural networks *in the worst-case* and *without* distributional assumptions, because we usually train our networks on arbitrary data and we are not a priori aware of simplifying structure in the data that could be exploited. In the worst-case setting, where arbitrary data in arbitrary dimension $d \geq 2$ is allowed, a lower bound of $\Omega(n)$ has been shown in (Munteanu et al., 2022), which is larger than the aforementioned $O(n/d)$ upper bounds under various assumptions. On the positive side, the $m = O(\lambda^{-4}n^4)$ worst-case bound of (Song and Yang, 2019) was improved to $m = O(\lambda^{-2}n^2)$ by combining their analysis with a coupled initialization (Munteanu et al., 2022). The gap between linear and quadratic was left as an open problem. To our knowledge, it is still the state of the art for distribution-independent worst-case bounds. A recent work (Karhadkar et al., 2024) studied eigenvalues of the NTK kernel matrix at initialization and also ran into a linear vs. quadratic gap.

The above works focused on general convex or specifically on squared loss which is standard for regression tasks. This was complemented by the logistic loss for binary classification. Separability assumptions for two-layer ReLU networks and smooth loss functions led to polynomial dependencies of the width $m$ on $n$ in (Cao and Gu, 2019b;a; Allen-Zhu et al., 2019a; Nitanda et al., 2019). A breakthrough result (Ji and Telgarsky, 2020) established a polylogarithmic dependence on $n$. In this under-parameterized regime for classification, a parameter $\gamma$ captures the maximum classification margin of the NTK. The upper bound of $m = O(\gamma^{-8}\log n)$ was complemented by a lower bound of $m = \Omega(\gamma^{1/2})$ against NTK analyses in (Ji and Telgarsky, 2020). Combining with the coupled initialization technique, (Munteanu et al., 2022) improved the upper bound to $m = O(\gamma^{-2}\log n)$ and corroborated tightness by a $\Omega(\gamma^{-2})$ lower bound against the standard initialization analysis. The general lower bound was improved to an *unconditional* $m = \Omega(\gamma^{-1})$ and $m = \Omega(\gamma^{-1}\log n)$ was established against NTK analyses. Generalization errors with SGD and gradient flow with quadratic $O(\gamma^{-2})$ dependence followed shortly after by (Telgarsky, 2022) using slightly different analysis methods, that allow for more movement than typical NTK analyses. To the best of our knowledge, the bounds of (Munteanu et al., 2022; Telgarsky, 2022) are the current state of the art in the worst-case without distributional or geometric data assumptions, and the quadratic vs. linear gap is an unresolved open problem in this regime.

Early convex formulations of neural networks are due to (Bengio et al., 2005) who leverage convexity in the measure space to develop a training algorithm that iteratively adds neurons. (Bach, 2017) leverage convexity to show that infinitely wide neural networks can break the curse of dimensionality. More recently, (Pilanci and Ergen, 2020) developed a finite-width convex reformulation for two-layer ReLU networks using duality theory. Their approach is based on enumerating all activation patterns encoded in cones. A significant body of work has developed ever since, including extensions to vector outputs (Sahiner et al., 2021), polynomial activations (Bartan and Pilanci, 2023), threshold activations (Ergen et al., 2023), and constrained optimization (Prakhya et al., 2025). Most relevant are (Mishkin et al., 2022; Dwaraknath et al., 2023) that draw connections to the NTK.

The notion of *convex neural networks* instead of the well-known *gated neural networks* is used to emphasize the property of allowing for convex training, which is of uttermost importance to our work. However, we note that two-layer gated ReLU neural networks denote the same family of networks as our convex formulation. Decoupling activation from the linear mapping was first

proposed by (Fiat et al., 2019), who called it *gated linear unit*. Their architecture is different from ours because they do not merge parameters on the two layers. As a result, the optimization problem is non-convex. (Mishkin et al., 2022) merges the parameters to obtain a convex model that they call *gated ReLU network*. They study connections between gated ReLU networks and standard ReLU networks based on convex cone programs and duality theory. Our work follows a different, more direct approach. We focus on gradient descent analysis, the explicit functional network formulations and their geometric interpretation. While we focus on two-layer ReLU networks, (Chen et al., 2021; Bartan and Pilanci, 2023) extend to multi-layer networks with ReLU activation. Furthermore, (Fiat et al., 2019) provide experiments showing that gated networks perform similar to standard neural networks even for small width and (Mishkin et al., 2022) compares different optimization approaches for convex neural networks.

Convex two-layer networks have remarkable similarities to random feature models (RFMs), (Rahimi and Recht, 2007; Rudi and Rosasco, 2017). Crucially, only the output weights of RFMs are trainable which considerably limits their expressibility (Yehudai and Shamir, 2019; Gonon, 2023).

## 2 PRELIMINARIES ON NEURAL NETWORKS

**Two-layer ReLU networks**   A two-layer ReLU network consists of a set of weights $w_1, \ldots, w_m \in \mathbb{R}^d$ for the first layer and weights $a_1, \ldots, a_m \in \{-1, 1\}$ for the second layer. These can be summarized as $(W, a) \in \mathbb{R}^{m \times d} \times \{-1, 1\}^m$. We will also use the notion of a weighted two-layer ReLU network $(W, a, \rho)$ if we have an additional weight vector $\rho \in [0, 1]^m$. We will usually assume that $\sum_{j=1}^m \rho_j = 1$, and in particular the unweighted case is the special case where we set all $\rho_j = \frac{1}{m}$ and omit $\rho$ from the notation for simplicity. We will refer to $m$ as the width of the network, which is an important parameter in the context of convergence analyses of gradient descent based neural network training. The classification of a point $x \in \mathbb{R}^d$ by the network $(W, a, \rho)$ is then given by

$$f_S(W, a, \rho, x) = \sum_{j=1}^m \rho_j a_j \langle x, w_j \rangle \mathbf{1}[\langle x, w_j \rangle > 0],$$

where $\mathbf{1}[r > 0] = 1$ if $r > 0$ and 0 otherwise. This simplifies by omitting $\rho$ in the uniform case.

For training and evaluating a network, we are given $n$ data points $x_1, \ldots, x_n \in \mathbb{R}^d$ and binary labels $y_1, \ldots, y_n \in \{-1, 1\}$ or real-valued targets $y_1, \ldots, y_n \in \mathbb{R}$ depending on the task at hand. We adopt the common standard assumptions that $\|x_i\| = 1$ and $|y_i| \in O(1)$ for all $i \in [n]$. We note that there are different normalizations in the literature. For instance a common normalization (Du et al., 2019c; Song and Yang, 2019; Ji and Telgarsky, 2020) is given by $1/\sqrt{m}$. We normalize by $1/m$ unless stated otherwise, which has the same effect if every weight is rescaled by $\sqrt{m}$ since it cancels the additional $1/\sqrt{m}$ factor. This simplifies our analyses and is equivalent, see Appendix A.

We specify a loss function $L_S(W) = \sum_{i=1}^n \ell(y_i, f(W, a, x_i))$, as a sum of individual losses. We will focus on the following choices. The logistic loss is often used for binary classification and is defined as $\ell(v_1, v_2) = \ln(1 + \exp(-v_1 v_2))$. The squared loss is given by $\ell(v_1, v_2) = \frac{1}{2}(v_1 - v_2)^2$ and is often used for regression with continuous target. Note that both are convex. Most of our analyses hold for arbitrary convex losses and it will be made clear when we focus on logistic loss.

**Convex two-layer ReLU networks**   In the infinite width limit, NTK theory (Jacot et al., 2018) ensures a stationary kernel and convexity of the training problem. This implies that gradient descent in the functional space converges to a globally optimal zero-error solution. Most, if not all, theoretical convergence results on gradient descent for training finite width two-layer ReLU networks (Du et al., 2019c; Song and Yang, 2019; Ji and Telgarsky, 2020) use the structural property that for *almost all* data points $x_i, i \in [n]$ and weight vectors $w_j, j \in [m]$, the activation of neurons, i.e., the indicator $\mathbf{1}[\langle x_i, w_j \rangle > 0]$ does not change during optimization. We note that this is the only source violating the convexity of the overall loss function $L_S$, given that the individual loss function $\ell$ is convex.

This motivates us to analyze a variant that we call *convex two-layer ReLU networks*, also known as *gated ReLU networks* (Fiat et al., 2019; Mishkin et al., 2022), where the activation is changed to stay constant after an initial random initialization. In addition to the set of weight vectors $w_1, \ldots, w_m \in \mathbb{R}^d$, we use a set $v_1, \ldots, v_m \in \mathbb{R}^d$ of orientation vectors that control the activation of neurons independently of the current choice of the corresponding weight vectors $w_i$. The parameterization

of such a convex network will be summarized as a pair of two matrices $(V, W) \in \mathbb{R}^{m \times d} \times \mathbb{R}^{m \times d}$. The classification of a point $x \in \mathbb{R}^d$ is then given by

$$f(V, W, \rho, x) = \sum_{j=1}^{m} \rho_j \langle x, w_j \rangle \mathbf{1}[\langle x, v_j \rangle > 0],$$

for an additional vector $\rho \in [0, 1]^m, \|\rho\|_1 = 1$ weighting the contributions of neurons. We omit $\rho$ in the uniform case as before. Due to the more complex activation function, we also do not need the sign vector $a$ for the second layer, or equivalently set $a_j = 1$ for all $j \in m$.

We initialize all weight vectors in $W$ to be $0$. The orientation vectors $V$ are initialized as usual for standard networks by drawing i.i.d. standard Gaussians. Crucially, the activations determined by $V$ do not change during optimization after initialization. Only the weights $W$ are optimized using a standard gradient descent update rule that iteratively minimizes the loss function $L(W) = L(V, W) = \sum_{i=1}^{n} \ell(y_i, f(V, W, x_i))$, where $\ell$ is an individual convex loss function as before.

Note that, in contrast to $L_S$, if $\ell$ is convex then $L(W)$ is also convex, since for any fixed choice of $V \in \mathbb{R}^{m \times d}$, any two weight matrices $W, W' \in \mathbb{R}^{m \times d}$ and any $t \in [0, 1]$, it holds that

$$L(tW + (1 - t)W')$$

$$= \sum_{i=1}^{n} \ell(y_i, f(V, tW + (1-t)W', x_i)) = \sum_{i=1}^{n} \ell\Big(y_i, \frac{1}{m} \sum_{j=1}^{m} \langle x_i, tw_j + (1-t)w'_j \rangle \mathbf{1}[\langle x, v_j \rangle > 0]\Big)$$

$$= \sum_{i=1}^{n} \ell\Big(y_i, \frac{t}{m} \sum_{j=1}^{m} \langle x_i, w_j \rangle \mathbf{1}[\langle x, v_j \rangle > 0] + \frac{(1-t)}{m} \sum_{j=1}^{m} \langle x_i, w'_j \rangle \mathbf{1}[\langle x, v_j \rangle > 0]\Big)$$

$$\leq \sum_{i=1}^{n} t\ell\Big(y_i, \frac{1}{m} \sum_{j=1}^{m} \langle x_i, w_j \rangle \mathbf{1}[\langle x, v_j \rangle > 0]\Big) + (1-t)\ell\Big(y_i, \frac{1}{m} \sum_{j=1}^{m} \langle x_i, w'_j \rangle \mathbf{1}[\langle x, v_j \rangle > 0]\Big)$$

$$= tL(W) + (1 - t)L(W').$$

By fixing the orientation vectors $V$, we thus remove the issue where the activation causes a non-convex structure in the standard version of two-layer ReLU networks. We also note that not all convex two-layer ReLU networks can be represented by regular two-layer ReLU networks. However, we will show that they behave very similarly to two-layer ReLU networks under mild standard conditions that were used *in previous literature* to derive convergence results.

**Motivation of convex two-layer ReLU networks** Our motivation for theoretically analyzing convex two-layer ReLU networks is to show that under mild conditions they are almost equivalent to the standard setting. Due to convexity, standard convergence analyses apply after successful initialization. By the equivalence they considerably simplify the theoretical analysis of standard two-layer ReLU neural networks regarding their width by reducing the analysis solely to the initialization. Further, our aim is to gain deeper insights into the role of the orientation and activation vectors for neural networks at initialization. This allows us to considerably strengthen results on the remaining linear vs. quadratic gaps on the width of two-layer ReLU networks that could not be resolved with previous approaches of (Munteanu et al., 2022; Karhadkar et al., 2024).

**Advantages** Convex two-layer ReLU networks allow a considerably simpler theoretical analysis since under any convex loss function their optimization remains convex. This is achieved by decoupling the weights that are optimized from the weights that control the activation of hidden neurons. Further, every standard two-layer ReLU network $(W, a)$ can be simulated by a convex two-layer ReLU network $(V, W')$ that yields the same classification function for all $x \in \mathbb{R}^d$. The opposite direction is not true, since there exist datasets that can be classified correctly by convex two-layer ReLU networks but cannot be classified correctly by any standard two-layer ReLU network, see Theorem 7.3. But under mild conditions on the relationship between data and orientation vectors $V$, a convex two-layer ReLU network $(V, W)$ of width $m$ can be simulated by a standard two-layer ReLU network $(W', a)$ of width $2m$ that yields the same classification function on the input data.

**Disadvantages** Convex two-layer ReLU networks require twice the memory of standard two-layer ReLU networks since we have to store two instead of just one vector for each hidden neuron. We note that the degrees of freedom remain the same since only one set of parameters is optimized. Further, they heavily depend on the initial choice of orientations that stay fixed during optimization.

We note that the latter disadvantage is not very restrictive. While standard two-layer ReLU networks adapt their weights and activations simultaneously, most, if not all theoretical analyses require that in fact almost all their activations *do not* change during optimization (Du et al., 2019c; Song and Yang, 2019; Ji and Telgarsky, 2020). Available initializations ensure desirable properties with high probability. Many in-depth analyses thus focus on properties of a successful initialization, and one can assume that the subsequent optimization converges, cf. (Karhadkar et al., 2024).

## 2.1 CONTRIBUTIONS AND ROADMAP

We state our three main contributions as follows:

1) We give an algorithm based on gradient descent on convex neural networks (NNs) that approximates the margin parameter $\gamma$ of standard non-convex NNs. Evaluating or approximating $\gamma$ was only known to be possible before in a few analytically tractable cases (Ji and Telgarsky, 2020; Munteanu et al., 2022; Telgarsky, 2022).

2) We improve the quadratic $m = \Omega(\gamma^{-2})$ worst case width lower bound of (Munteanu et al., 2022) to not only hold against the standard perfect NTK separator, but against *any* perfect NTK separator that is not adaptive to the initial weights. This new lower bound thus holds against all upper bound analyses since adaptivity has not yet been explored.

3) Along the way, we analyze and establish novel theoretical properties of convex (resp. gated) two-layer ReLU neural networks, in particular their close connection to standard two-layer ReLU neural networks and their NTK properties.

All missing details and formal proofs are in the appendix. The rest of our paper is structured as follows:

• In Section 3 we define a data dependent set of cones $S_0(X)$. We use the cones to show that the convex and standard variants of two-layer ReLU networks are almost equivalent. We also show that any (possibly infinitely wide) network is equivalent to a network of finite width at most $|S_0(X)| = O(n^{d-1})$.

• In Section 4, we study two common parameters, the NTK separation margin $\gamma$ and the smallest eigenvalue $\lambda$ of the NTK kernel matrix. We show that there exist weighted two-layer ReLU networks of width at most $|S_0|$ for which the infinite width limits of these parameters are attained.

• In Section 5, we show that for the networks studied in Section 3 the gradient and weight updates are similar as long as the weights of standard networks do not change neuron activations too much.

• In Section 6, we consider gradient descent for optimizing convex two-layer ReLU networks of small width with logistic loss for binary classification. In particular, we show that standard gradient descent converges to a network $(V, W)$ that satisfies $(1 + \varepsilon)\gamma \geq \min_{i \in [n]} y_i f(V, W/\max_{j \in [m]} \|w_j\|_2, x_i) \geq (1 - \varepsilon)\gamma$, thus providing a provable approximation algorithm for calculating $\gamma$.

• In Section 7, we consider two datasets: 1) we study the *alternating points on the circle* data to show that existing non-adaptive techniques for proving $m = O(\gamma^{-2})$ (omitting parameters other than $\gamma$) cannot give a bound of $O(\gamma^{-(2-\delta)})$, for any $\delta > 0$, as we show that any perfect NTK separator mapping must be chosen *adaptively* to the initial weights unless $m = \Omega(\gamma^{-2})$.

2) we study the *three-dimensional hypercube with parity labels* data and show that convex two-layer ReLU networks can perfectly classify this data using orientations that are orthogonal to data points, while any standard two-layer ReLU network must have at least one misclassification.

## 3 CONES AND EQUIVALENCE OF CONVEX AND STANDARD NETWORKS

The following lemma shows that the two variants of neural networks are very similar in the sense that standard ReLU networks can be simulated by convex ReLU networks such that all points in the dataset evaluate to the same classification (resp. target value). The reverse simulation is also possible albeit under a factor two blow-up of the width and under a mild condition on the relationship between data and orientation vectors $V$. Related, though different uni-directional equivalences appeared in (Mishkin et al., 2022; Pilanci and Ergen, 2020; Mishkin and Pilanci, 2023). Our result was previously unknown and establishes a bi-directional equivalence.

**Theorem 3.1.** *For any two-layer ReLU network $(W', a) \in \mathbb{R}^{m \times d} \times \{-1, 1\}^m$ there exists a convex two-layer ReLU network $(V, W) \in \mathbb{R}^{m \times d} \times \mathbb{R}^{m \times d}$ such that for all $x \in \mathbb{R}^d$ it holds that $f(V, W, x) = f_S(W', a, x)$. Further, for any convex two-layer ReLU network $(V, W) \in \mathbb{R}^{m \times d} \times \mathbb{R}^{m \times d}$ such that for any $i \in [n], j \in [m]$ it holds that $\langle x_i, v_j \rangle \neq 0$, there exists a two-layer ReLU network $(W', a) \in \mathbb{R}^{2m \times d} \times \{-1, 1\}^{2m}$ such that for any $i \in [n]$ it holds that $f_S(W', a, x_i) = f(V, W, x_i)$.*

Note that for the first transformation, the number of hidden neurons stays the same. But in the other direction the number grows from $m$ to $2m$. Also note that in the set of functions that standard networks represent, can also be represented by convex networks. In particular this implies that the ability of generalization is at least preserved. In the other direction, the equivalence of functions is restricted to training data, which is due to higher expressibility of convex networks, cf. Theorem 7.3.

Most of our analysis will be centered around the following data-dependent definition of cones, which is originally due to (Munteanu et al., 2022). We define for any subset $U \subseteq [n]$ a cone
$$C(U) = C(U, X) = \{x \in \mathbb{R}^d \mid \langle x, x_i \rangle > 0 \text{ iff } i \in U\}.$$
We remark that the disjoint union of all cones satisfies $\dot{\bigcup}_{U \subseteq [n]} C(U) = \mathbb{R}^d$ and it holds that $C(\emptyset) = \{x \in \mathbb{R}^d \mid \langle x, x_i \rangle \leq 0 \text{ for all } i \in [n]\}$. We set $S_0 := S_0(X) = \{C(U) \mid U \subseteq [n], C(U) \neq \emptyset\}$. By definition we have $|S_0| \leq 2^n$, but as a direct consequence of Theorem 1 in (Cover, 1965) it follows that $|S_0|$ is actually bounded by $O(n^{d-1})$. We include a proof in the appendix for completeness.

**Lemma 3.2.** *For any dataset $X$ it holds that $|S_0(X)| \leq 4n^{d-1}$. Further if $X$ is in general position and $n \geq d > 2$, i.e., any subset of $d$ points is linearly independent, then $|S_0(X) \setminus \{0\}| = \sum_{k=0}^{d-1} \binom{n}{k}$.*

The following lemmas are novel and show that if our dataset is finite, then for every (convex) two-layer ReLU network there exists a (convex) two-layer ReLU network of width at most $|S_0| = O(n^{d-1})$ such that their classification is the same for all $x_i, i \in [n]$. The idea is that in fact we need only one orientation vector in each cone that is hit by at least one orientation in the original network, which reduces their remaining analysis to the set of cones in $S_0$ rather than all orientation vectors.

**Lemma 3.3.** *For any convex two-layer ReLU network $(V, W) \in \mathbb{R}^{m \times d} \times \mathbb{R}^{m \times d}$ let $S_V = \{C \in S_0 \mid \exists j \in [m] : v_j \in C\}$ and $m' = |S_V| \leq \min\{m, |S_0|\}$. Then there exists a convex two-layer ReLU network $(V', W') \in \mathbb{R}^{m' \times d} \times \mathbb{R}^{m' \times d}$ together with weights $\rho_1, \ldots, \rho_{m'}$ such that for all $i \in [n]$ it holds that*

$$f(V, W, x_i) = \frac{1}{m} \sum_{j=1}^{m} \langle x_i, w_j \rangle \mathbf{1}[\langle x_i, v_j \rangle > 0] = \sum_{j=1}^{m'} \rho_j \langle x_i, w'_j \rangle \mathbf{1}[\langle x_i, v'_j \rangle > 0] = f(V', W', \rho, x_i).$$

We note that the weights are not necessary to obtain the result as we can replace $w'_j$ by $w'_j \cdot \rho_j$. However, if we take the derivative with respect to $w_j$, using the weighted version simplifies the presentation and allows us to argue that the gradient also remains the same.

We have an equivalent novel result for standard two-layer networks. However, up to technical details including the weights, it also follows if we transform the standard network to a convex network by Theorem 3.1, then adjust (reduce) its size using Lemma 3.3 and then apply Theorem 3.1 again to get the equivalent standard two-layer network.

**Lemma 3.4.** *For any two-layer ReLU network $(W, a) \in \mathbb{R}^{m \times d} \times \{-1, 1\}^m$ let $S_V = \{(C, a_0) \in S_0 \times \{-1, 1\} \mid \exists j \in [m] : w_j \in C\}$ and $m' = |S_V| \leq \min\{m, 2|S_0|\}$. Then there exists a two-layer ReLU network $(W', a') \in \mathbb{R}^{m' \times d} \times \mathbb{R}^{m' \times d}$ together with weights $\rho_1, \ldots, \rho_{m'}$ such that for all $x_i, i \in [n]$ it holds that*

$$f_S(W, a, x_i) = \frac{1}{m} \sum_{j=1}^{m} a_j \langle x_i, w_j \rangle \mathbf{1}[\langle x_i, w_j \rangle > 0]$$
$$= \sum_{j=1}^{m'} \rho_j \langle x_i, w'_j \rangle \mathbf{1}[\langle x_i, w'_j \rangle > 0] = f_S(W', a', \rho, x_i).$$

## 4 SEPARATION MARGIN AND THE SMALLEST EIGENVALUE OF THE NTK

In this section, we define our versions of the two parameters that are used to bound the width of two-layer ReLU networks for binary classification with logistic loss and regression with squared

loss. We first define the smallest eigenvalue of the NTK $\lambda$ introduced in (Du et al., 2019c). We note that $\lambda$ is tightly related to separation and collinearity conditions studied earlier, e.g. in (Li and Liang, 2018; Oymak and Soltanolkotabi, 2020).

**Smallest eigenvalue of the NTK kernel matrix** The NTK kernel matrix $H \in \mathbb{R}^{n \times n}$ is defined as in previous work by

$$H_{ij} = \mathbb{E}_{v \sim \mathcal{N}(0,I)}[\langle x_i, x_j \rangle \mathbf{1}[\langle x_i, v \rangle > 0, \langle x_j, v \rangle > 0]]$$

We set $\lambda = \lambda(X) = \lambda(H)$ to be the minimum eigenvalue of $H$.

Given a matrix of activation vectors $V$, and a vector of weights $\rho$, we further define the finite counterpart. To this end, recall that the default is $\rho_k = 1/m$, which corresponds to previous work.

$$H_{ij}^{\mathrm{dis}} = H_{ij}^{\mathrm{dis}}(V) = \sum_{k=1}^m \rho_k \langle x_i, x_j \rangle \mathbf{1}[\langle x_i, v_k \rangle > 0, \langle x_j, v_k \rangle > 0]$$

and $\lambda_V = \lambda(X, V) = \lambda(H^{\mathrm{dis}})$ to be the minimum eigenvalue of $H^{\mathrm{dis}}$.

**NTK separation margin** Next, we define the NTK separation margin parameter $\gamma$, which was introduced in (Ji and Telgarsky, 2020) and further analyzed in (Munteanu et al., 2022). Intuitively, $\gamma$ quantifies the maximum classification margin of the points in the RKHS of the NTK. Let $B = B^d = \{x \in \mathbb{R}^d \mid \|x\|_2 \leq 1\}$ be the unit ball in $d$ dimensions. We set $\mathcal{F}_B$ to be the set of functions $f$ mapping from $\mathrm{dom}(f) = \mathbb{R}^d$ to $\mathrm{range}(f) = B$. Let $\mu_\mathcal{N}$ denote the Gaussian measure on $\mathbb{R}^d$, specified by the Gaussian density with respect to the Lebesgue measure on $\mathbb{R}^d$.

**Definition 4.1.** Given a data set $(X, Y) \in \mathbb{R}^{n \times d} \times \mathbb{R}^n$ and a map $\overline{v} \in \mathcal{F}_B$ we set $\gamma_{\overline{v}}$ equal to

$$\gamma_{\overline{v}}(X, Y) := \min_{i \in [n]} y_i \int \langle \overline{v}(z), x_i \rangle \mathbf{1}[\langle x_i, z \rangle > 0] \, d\mu_\mathcal{N}(z).$$

Further set $\gamma = \gamma(X, Y) := \max_{\overline{v}' \in \mathcal{F}_B} \gamma_{\overline{v}'}$ We say that $\overline{v}$ is optimal if $\gamma_{\overline{v}} = \gamma$.

We note that $\max_{\overline{v}' \in \mathcal{F}_B} \gamma_{\overline{v}'}$ always exists since $\mathcal{F}_B$ is a set of bounded functions on a compact subset of $\mathbb{R}^d$. In (Munteanu et al., 2022) it was shown that for every mapping $\overline{v}$, there exists another mapping $\overline{v}'$, which is constant on all cones in $S_0$ and satisfies that $\gamma_{\overline{v}'} = \gamma_{\overline{v}}$. This implies that there exists a finite (weighted) convex neural network that satisfies $y_i f(V, W, x_i) \geq \gamma_{\overline{v}'}$. In particular it holds that equality is attained for every optimal $\overline{v}$.

The network is given by $m = |S_0|$, $V$ consists of one representative $v_C \in C$ for each cone $C \in S_0$, $w_C = \overline{v}'(v_C)$ and $\rho_C = P(z \in C)$ according to the Gaussian measure as in Definition 4.1.

Given $V \in \mathbb{R}^{m \times d}$, and any $B \in \mathbb{R}^{>0}$, we set $\mathcal{W}_B = \{W \in \mathbb{R}^{m \times d} \mid \|w_j\|_2 \leq B \text{ for all } j \in [m]\}$ and define $\gamma_V = \max_{W \in \mathcal{W}_1} \min_{i \in [n]} y_i f(V, W, x_i, \rho)$. Moreover, we set $W_V \in \mathcal{W}_1$ to be a weight matrix that attains the maximum, i.e., it holds that

$$\gamma_V = \min_{i \in [n]} y_i f(V, W_V, x_i). \tag{1}$$

**Convex two-layer networks can attain the infinite width limit parameters** To show that gradient descent applied to two-layer ReLU networks converges, one usually shows that if the width $m$ of the network is large enough, then the finite width parameters $\gamma_W$ resp. $\lambda_W$ for the initial weight vectors $W_0$ are close to their infinite width limits $\gamma$ resp. $\lambda$. Further one shows that this does not change significantly during the iterative optimization procedure.

We note that for convex two-layer ReLU networks, the values of $\gamma_V$ resp. $\lambda_V$ are determined at initialization since they depend only on $V$ and the data, both of which do not change during optimization. The parameters thus do not change at all which makes the second argument obsolete and simplifies the convergence analysis. The following theorem establishes that there exists a weighted convex network $(V, W_V, \rho)$ such that $\gamma_V = \gamma$ and $\lambda_V = \lambda$. This novel finding will be important for provably approximating the (theoretical) integral valued quantity $\gamma$ in Section 6 using a convex network and a simple gradient descent.

**Theorem 4.2.** *Let $m = |S_0|$ and for $C \in S_0$ set $\rho_C = P(C)$ where $P(C)$ is the probability that a random vector in $V$ is in $C$. Further let $v_C$ be any vector in $C$, and let $V$ be the matrix whose rows are the collection of all $v_C, C \in S_0$. Then it holds that $\lambda_V = \lambda$ and $\gamma_V = \gamma$.*

## 5   GRADIENT DESCENT WEIGHT UPDATES

In this section, we study the loss function and its directional derivatives showing more similarities between the two variants of ReLU networks. Further, we show that the gradient behaves similarly in both formulations of two-layer ReLU networks, which prepares the subsequent convergence results presented in the next section. For a weighted convex neural network $(V, W, \rho)$ we set $L'(W)$ to be the gradient of the loss function $L(W)$, i.e.,

$$L'(V, W, \rho)_j = \frac{\partial L(W)}{\partial w_j} = \sum_{i \in [n], \langle x_i, v_j \rangle > 0} \partial_{w_j} \ell(y_i, f(V, W, \rho, x_i)).$$

We note that

$$L'(V, W, \rho) = \sum_{i=1}^{n} \ell'(y_i, f(V, W, \rho, x_i)) \nabla f(V, W, \rho, x_i),$$

where

$$(\nabla f(V, W, \rho, x_i))_j = \frac{\partial f(V, W, \rho, x_i)}{\partial w_j} = \rho_j \mathbf{1}[\langle x_i, v_j \rangle > 0] x_i.$$

Thus, we also have for any $W, W' \in \mathbb{R}^{m \times d}$ that

$$\langle \nabla f(V, W, \rho, x_i), W' \rangle = \sum_{j \in [m]} \langle \nabla f(V, W, \rho, x_i))_j, w'_j \rangle = f(V, W', \rho, x_i).$$

Similarly, we have for a weighted standard ReLU network $(W, a, \rho)$ that

$$L'_S(W, a, \rho)_j = \frac{\partial L_S(W)}{\partial w_j} = \sum_{i \in [n], \langle x_i, w_j \rangle > 0} \partial_{w_j} \ell(y_i, f_S(W, a, \rho, x_i)),$$

$$L'_S(W, a, \rho) = \sum_{i=1}^{n} \ell'(y_i, f_S(W, a, \rho, x_i)) \nabla f(W, a, \rho, x_i),$$

$$(\nabla f(W, a, \rho, x_i))_j = \frac{\partial f_S(W, a, \rho, x_i)}{\partial w_j} = \rho_j a_j \mathbf{1}[\langle x_i, w_j \rangle > 0] x_i,$$

and $\langle \nabla f(W', a, \rho, x_i), W' \rangle = f_S(W', a, \rho, x_i)$.

Recall that the default vector $\rho$ has all entries equal to $1/m$. In the following, we consider the equivalent networks from previous sections and show that their gradient based weight updates have a similar effect on all variants. More precisely, we start with a standard two-layer ReLU network $(W, a)$. We then define a transformation map $T_1$, which maps $(W, a)$ to the equivalent network $(V, W')$ as in Theorem 3.1, $T_2$, which maps $(W, a)$ to the network $(V', W'', \rho)$ which is similar to $(V, W')$ by Lemma 3.3, and $T_3$, which maps $(W, a)$ to the network $(W', a', \rho')$ as in Lemma 3.4. The exact formal definitions of $T_1, T_2$, and $T_3$ are detailed in Appendix D. They are technically slightly different in that they only map the matrices to one another, not the whole networks. But their idea follows along the lines of the intuitive explanation above. They are used in the following lemma to express that the gradients of transformed networks equal the transforms of the gradients.

**Lemma 5.1.** *For the gradient it holds that*

$$L'(V, T_1(W, a)) = T_1(L'(V, W), a) \tag{2}$$

$$L'(V', T_2(W, a), \rho) = T_2(L'(V, W), a) \tag{3}$$

$$L'_S(T_3(W, a), a', \rho') = T_3(L'(V', W, \rho'), a). \tag{4}$$

*Further for any weight update $\Delta W$ such that for all $i \in [n]$ it holds that $\mathbf{1}[\langle x_i, w_j \rangle > 0] = \mathbf{1}[\langle x_i, w_j + \Delta w_j \rangle > 0]$ we have that*

$$L_S(W + \Delta W, a) = L(V, T_1(W + \Delta W, a))$$
$$= L(V', T_2(W + \Delta W, a), \rho) = L_S(T_3(W + \Delta W, a), a', \rho'). \tag{5}$$

The lemma thus proves that the network transformations that apply to the weight matrices of equivalent networks, also apply to the matrices that carry all partial derivatives. Thus weight updates have a similar effect across all network types. This is again a novel and important reduction, which implies that gradient descent has similar training dynamics on convex networks as on non-convex standard networks under their usual properties and thus enables our approximation results covered in the next section.

# 6 GRADIENT DESCENT APPROXIMATION RESULTS

Next, we establish an approximation result for convex two-layer ReLU networks with logistic loss, i.e., $\ell(r) = \ln(1 + \exp(-r))$. Recall that we initialize $V$ according to a suitable distribution such as i.i.d. Gaussians and set $W = W_0$ to be a zero matrix. We keep $V$ fixed during training and apply gradient descent to update the weights $W_t$ for $t \geq 0$ in an iterative manner

$$W_{t+1} = W_t - \eta_t L'(W_t)$$

where $\eta_t \in \mathbb{R}_{\geq 0}$ is a learning rate parameter and $L'(W_t)$ is the gradient of the loss function $L(W_t)$ at $W_t$, i.e.,

$$L'(W_t)_j = \frac{\partial L(W_t)}{\partial (W_t)_j} = \sum_{i \in [n], \langle v_j, x_i \rangle > 0} x_j \partial_{w_j} \ell(y_i, f(V, W_t, x_i)).$$

We note that this reaches a factor 2 approximation to the optimal solution using standard gradient descent analyses (Nesterov, 2004; Bubeck, 2015) in $O(B^2)$ iterations by simple boundedness and Lipschitz arguments detailed in the appendix. The following lemma shows that after gradient descent converges to a constant factor approximation, it is a $(1 \pm \varepsilon)$-approximation of the real value of $\gamma_V$.

**Lemma 6.1.** *Let $\gamma_V > 0$ as defined in Equation (1). Let $\ell(r) = \ln(1 + \exp(-r))$ and $\varepsilon > 0$. Let $B \geq (\ln(4) + \ln(n))/(\varepsilon \gamma_V)$. Let $W^* \in \mathcal{W}_B$ minimize $L(W)$ and let $W \in \mathcal{W}_B$ be any solution such that $L(W) \leq 2L(W^*)$. Then it holds that $\gamma_V \geq \min_{i \in [n]} y_i f(V, W/B, x_i) \geq (1 - \varepsilon)\gamma_V$*

We note that the previous Lemma has a circular dependence on the value of $\gamma_V$ because to estimate this quantity, we need to find the right value of $B$, which depends on $\gamma_V$ again. This can be handled simply by guessing $\gamma_V$ in powers of 2, which contributes only an additional factor of $\log_2(1/\gamma_V)$ to the number of iterations.

Finally, the following theorem shows how $\gamma_V$, approximated by gradient descent in Lemma 6.1, can be related to the infinite width limit $\gamma$ up to a $(1 \pm \varepsilon)$ multiplicative error.

**Theorem 6.2.** *Assume $V \in \mathbb{R}^{m \times d}$ is initialized with i.i.d. Gaussians and $0 < \delta \leq \varepsilon$. For any $m$ it holds that $\mathbb{E}\gamma_V \leq \gamma$ over the Gaussian measure. Further, if the network width is $m \geq c \cdot (\varepsilon \delta \gamma)^{-2} \ln(n/\varepsilon)$ for an absolute constant $c > 0$, then with probability at least $1 - \delta$ it holds that $(1 + \varepsilon)\gamma \geq \gamma_V \geq (1 - \varepsilon)\gamma$.*

The Gaussian initialization of $V$ and a refinement by gradient descent that updates only the weights $W$ thus suffices to estimate the infinite width limit value of $\gamma$ up to arbitrary precision. This is an important main finding of our work, because evaluating or approximating the true infinite width limit value of $\gamma$ was only known in a few special and analytically tractable cases, see for instance (Ji and Telgarsky, 2020; Munteanu et al., 2022).

# 7 TWO IMPORTANT DATA EXAMPLES

**The alternating circle and why it might be hard to prove that a two-layer ReLU network of linear width suffices for arbitrarily small error** Consider the following set of points for $k \in [n]$:

$$x_k = \left( \cos\left(\frac{2k\pi}{n}\right), \sin\left(\frac{2k\pi}{n}\right) \right) \text{ and } y_k = (-1)^k.$$

The dataset consists of equidistant points on the circle with alternating labels. It has been used in (Munteanu et al., 2022) to derive lower bounds of different strengths on the width of two-layer ReLU networks. In particular we will strengthen their $\Omega(\gamma^{-2})$ bound to hold against *any fixed* choice of $\overline{v}$, not only for a special choice of $\overline{v}$ commonly used in (Ji and Telgarsky, 2020; Munteanu et al., 2022) for proving upper bounds. This implies that proving any upper bound better than $O(\gamma^{-2})$ requires choosing $\overline{v}$ *adaptively* to the initial weights, which to our knowledge is completely unexplored except for the case of two dimensional data, see Lemma 3.5 resp. F.2 (Munteanu et al., 2022), which allows a width of $O(\gamma^{-1} \log n)$. The authors state however, that the same construction is *not extendable* to higher dimensions, as in 3 dimensions or higher, the bounds achieved by their construction deteriorate to values larger than $O(\gamma^{-2})$.

We first prove the following technical lemma. It establishes that the data set has a small margin $\gamma \approx 1/n$ and for *any* NTK separator, the estimate for some data point must have high variance.

**Lemma 7.1.** *(informal version of Lemma F.2) Let* $X$ *be the* alternating points on the circle *dataset defined above with* $n \equiv 0 \mod 4$. *Then the following holds:*
*(1) The separation margin is given by* $\gamma_X = \Theta(1/n)$
*(2) For any fixed map* $\overline{v} \in \mathcal{F}_B$ *there exists an index* $i \in [n]$ *such that for an absolute constant* $c$

$$\frac{\int \langle \overline{v}(z), y_i x_i \rangle \mathbf{1}[\langle x_i, z \rangle > 0] \, d\mu_{\mathcal{N}}(z)}{\int |\langle \overline{v}(z), y_i x_i \rangle| \mathbf{1}[\langle \overline{v}(z), y_i x_i \rangle < 0] \mathbf{1}[\langle x_i, z \rangle > 0] \, d\mu_{\mathcal{N}}(z)} \leq c\gamma_X$$

*and for* $Z_i = \langle \overline{v}(z), y_i x_i \rangle \mathbf{1}[\langle x_i, z \rangle > 0]$ *we have that* $\mathrm{Var}(Z_i) \geq \Omega(\gamma_X^{-2} \mathbb{E}(Z_i)^2)$.

Using Lemma 7.1 we prove for our *alternating points on the circle* dataset $X$ that if the width of the network is $m = o(\gamma^{-2})$ then for any fixed perfect NTK separator, the finite network has with constant probability at least one misclassification.

**Theorem 7.2.** *Let* $X$ *be the* alternating points on the circle *dataset with* $n$ *divisible by* $4$ *and let* $W \in \mathbb{R}^{m \times 2}$ *be a matrix consisting of* $m$ *Gaussians. Then there is a constant* $c_0 > 0$ *such that if* $m \leq c_0 \gamma_X^{-2}$, *then for any fixed* $\overline{v} \in \mathcal{F}_B$ *there exists an index* $i \in [n]$ *such that with constant probability*

$$\frac{1}{m} \sum_{s=1}^{m} y_i \langle \overline{v}(w_s), x_i \rangle] \mathbf{1}[\langle x_i, w_s \rangle > 0] \leq 0.$$

Thus, our result reveals that constructing the perfect NTK separator $\overline{v}$ *adaptively* to the initialization is the only last hope for linear $\tilde{O}(\gamma^{-1})$ upper bounds (or anything between linear and quadratic) in the worst case setting. Our $\Omega(\gamma^{-2})$ lower bound thus almost closes an important open problem, since it matches the previous $O(\gamma^{-2})$ upper bounds of (Munteanu et al., 2022; Telgarsky, 2022) and adaptivity has never been explored in previous work except for the aforementioned case that is restricted to 2 dimensional data. This motivates studying adaptive techniques that extend to arbitrary dimensions as a future research direction.

**The 3-dimensional hypercube and cones of measure zero** The next example we want to consider is the 3-dimensional hypercube with parity labels. More precisely, the dataset is given by $X = \{-1, 1\}^3$ and for $x \in X$ we set $y_x = x_1 x_2 x_3$, i.e., $y_x = 1$ if the number of 1's in $x$ is odd, and otherwise we set $y_x = -1$. This toy example was studied before in (Munteanu et al., 2022). In our context it becomes important for the following new reason: we have that $\gamma_X = 0$ and we will show in the following that there exists no standard two-layer ReLU network that correctly classifies all points. However, there exists a *convex* two-layer ReLU network that classifies all points correctly using cones of measure zero. The following theorem thus highlights an important difference in the expressibility of standard compared to convex networks.

**Theorem 7.3.** *Let* $X$ *be the* 3-*dimensional hypercube with parity labels. Then the following holds:*
*(1)* $\gamma_X = 0$,
*(2) there exists no (standard) two-layer ReLU network that classifies all points correctly,*
*(3) there exists a convex two-layer ReLU network that classifies all points correctly.*

## 8 CONCLUSION

We theoretically analyzed *convex* two-layer ReLU networks, which are a strict generalization of the standard non-convex formulation with similar properties. Under mild assumptions that are standard in previous literature, we have shown that they are almost equivalent to standard two-layer ReLU networks. Their main purpose in our context is simplifying the theoretical analysis of two-layer ReLU networks that in their standard formulation require considerable technical overhead for controlling the amount of weights and data points, that change the activation of neurons during training. Using convex networks, we showed new properties that by equivalence extend to standard two-layer ReLU networks. Convex networks allow for standard gradient descent analyses to apply directly, based on which we showed how to approximate the NTK classification margin $\gamma$ up to a $(1 \pm \varepsilon)$ factor. We also strengthened existing quadratic lower bounds on the width, which imply that current analyses are tight and improving worst-case upper bounds below the $\Omega(\gamma^{-2})$ barrier requires currently unexplored adaptive techniques for constructing a perfect NTK separator during or after initialization. We hope our methods will be extended to yield better bounds on the width of two-layer ReLU networks in future research or finally lead to unconditional $\Omega(\gamma^{-2})$ lower bounds.

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

## A    EQUIVALENCE OF DEFINITIONS

In the standard literature we have that classification of a point is given by

$$f_0(W', a, x) = \frac{1}{\sqrt{m}} \sum_{j=1}^{m} a_j \langle x, w_j' \rangle \mathbf{1}[\langle x, w_j' \rangle > 0],$$

which results in a normalized classification if $W'$ is normalized by $1/\sqrt{m}$ as well. Further each gradient for logistic loss is 1-Lipschitz leading to a convergence when using a step size of 1 (resp. $1/n$). In our definition, the norm of each row of $W$ is independent of $m$ and thus can be rescaled by a factor $\sqrt{m}$ without problems. If we set the step size to $m$ (resp. $m/n$) then each step in the gradient descent has the exact same effect as it would have in the standard definition.

More precisely we get the following equivalences:

Let $W \in \mathbb{R}^{m \times d}$ be any matrix and set $W' = W/\sqrt{m}$. Further set $L_0(W') = \sum_{i \in [n], \langle x_i, w_j' \rangle > 0} \ell(y_i, f_0(W', a, x_i))$ and $L_0'(W') = \sum_{i \in [n]} \ell'(y_i, f_0(W', a, x_i)) \nabla f_0(W', a, x_i)$.

Then we have that

$$f_S(W, a, x) = \frac{1}{m} \sum_{j=1}^{m} a_j \langle x, w_j \rangle \mathbf{1}[\langle x, w_j \rangle > 0]$$

$$= \frac{1}{\sqrt{m}} \sum_{j=1}^{m} a_j \langle x, w_j' \rangle \mathbf{1}[\langle x, w_j' \rangle > 0]$$

$$= f_0(W', a, x).$$

Further it holds for all $j \in [m]$ that

$$(\nabla f_S(W, a, x))_j = \frac{1}{m} a_j \mathbf{1}[\langle x, w_j \rangle > 0] x$$

$$= \left( \frac{1}{\sqrt{m}} a_j \mathbf{1}[\langle x, w_j \rangle > 0] x \right) / \sqrt{m}$$

$$= \left( \frac{1}{\sqrt{m}} a_j \mathbf{1} \left[ \left\langle x, \frac{w_j}{\sqrt{m}} \right\rangle > 0 \right] x \right) / \sqrt{m}$$

$$= \frac{1}{\sqrt{m}} (\nabla f_0(W', a, x))_j.$$

Thus, for any $\eta \in \mathbb{R}$, we have that

$$\frac{1}{\sqrt{m}} \left( W - m\eta L_S'(W) \right) = \frac{W}{\sqrt{m}} - \frac{m}{\sqrt{m}} \eta \sum_{i \in [n]} \ell'(y_i, f_S(W, a, x_i)) \nabla f_S(W, a, x_i)$$

$$= \frac{W}{\sqrt{m}} - \frac{m}{\sqrt{m}} \eta \sum_{i \in [n]} \left( \ell'(y_i, f_0(W', a, x_i)) \right) \frac{1}{\sqrt{m}} \nabla f_0(W', a, x_i)$$

$$= W' - \eta \sum_{i \in [n]} \ell'(y_i, f_0(W', a, x_i)) \nabla f_0(W', a, x_i)$$

$$= W' - \eta L_0'(W').$$

## B    CONES AND EQUIVALENCE OF CONVEX AND STANDARD NETWORKS

Given a subset $U \subseteq [n]$ we define the following cone:

$$C(U) = C(U, X) = \{x \in \mathbb{R}^d \mid \langle x, x_i \rangle > 0 \text{ if and only if } i \in U\}.$$

Note that $C(\emptyset) = \{x \in \mathbb{R}^d \mid \langle x, x_i \rangle \leq 0 \text{ for all } i \in [n]\}$ and that the disjoint union of all cones satisfies $\dot{\bigcup}_{U \subseteq [n]} C(U) = \mathbb{R}^d$. For any cone $C = C(U)$ we set $U(C) = U$. Further we set $P(U)$ to be the probability that a random Gaussian is an element of $C(U)$ and $P_U$ to be the probability measure of random Gaussians $z \sim \mathcal{N}(0, I_d)$ restricted to the event that $z \in C(U)$, where $I_d \in \mathbb{R}^{d \times d}$ denotes the $d$ dimensional identity matrix.

Given a matrix $M \in \mathbb{R}^{m \times d}$ we set $K(M) = \{x \in \mathbb{R}^d \mid \exists j \in [m] : \langle m_j, x \rangle = 0\}$ to be the union of the hyperplanes that are orthogonal to one of the rows of $M$.

The following theorem shows that the two variants of neural networks are very similar in the sense that standard ReLU networks can be simulated by convex ReLU networks such that all points in the dataset evaluate to the same classification (resp. target value). The reverse simulation is also possible albeit under a factor two width blow-up and under a mild condition on the relationship between data and orientation vectors $V$.

**Theorem B.1.** *For any two-layer ReLU network $(W', a) \in \mathbb{R}^{m \times d} \times \{-1, 1\}^m$ there exists a convex two-layer ReLU network $(V, W) \in \mathbb{R}^{m \times d} \times \mathbb{R}^{m \times d}$ such that for all $x \in \mathbb{R}^d$ it holds that $f_S(W', a, x) = f(V, W, x)$. Further for any convex two-layer ReLU network $(V, W) \in \mathbb{R}^{m \times d} \times \mathbb{R}^{m \times d}$ with $K(X) \cap \{v_1, \ldots, v_m\} = \emptyset$ (i.e., for any $i \in [n], j \in [m]$ we have that $\langle x_i, v_j \rangle \neq 0$), there exists a two-layer ReLU network $(W', a) \in \mathbb{R}^{2m \times d} \times \{-1, 1\}^{2m}$ such that for any $i \in [n]$ we have that $f_S(W', a, x_i) = f(V, W, x_i)$.*

*Proof.* For the first part of the lemma we simply set $w_j = a_j w'_j$ and $v_j = w'_j$. Then it follows immediately for any $x \in \mathbb{R}^d$ that

$$f_S(W', a, x) = \frac{1}{m} \sum_{j=1}^{m} a_j \langle x, w'_j \rangle \mathbf{1}[\langle x, w'_j \rangle > 0]$$

$$= \frac{1}{m} \sum_{j=1}^{m} \langle x, a_j w'_j \rangle \mathbf{1}[\langle x, w'_j \rangle > 0]$$

$$= \frac{1}{m} \sum_{j=1}^{m} \langle x, w_j \rangle \mathbf{1}[\langle x, v_j \rangle > 0] = f(V, W, x).$$

For the second part note that the infimum $\alpha = \inf_{j \in [m], z \in K(X)} \{\|z - v_j\|_2\}$ is attained as it is the minimum distance of the finite set of data points and a finite set of hyperplanes that by assumption do not contain any of the input points and thus it must be strictly greater than 0. For $j \in [m]$ we set $w'_j = v_j \cdot \frac{2\|w_j\|_2}{\alpha} + w_j$, $a_j = 1$ and $w'_{j+m} = v_j \cdot \frac{2\|w_j\|_2}{\alpha}$ and $a_{j+m} = -1$. Note that for any $i \in [n]$ and $j \in [m]$ we have that

$$\mathbf{1}[\langle x_i, w'_j \rangle > 0] = \mathbf{1}[\langle x_i, v_j \rangle > 0]$$

as we also have that $\inf_{z \in K(X)} \{\|z - v_j \cdot \frac{\|w_j\|_2}{\alpha}\|_2\} \geq 2\|w_j\|_2$ and thus the sign of all points in $v_j \cdot \frac{2\|w_j\|_2}{\alpha} + \beta w_j$ for $\beta \in [0, 1]$ are the same.

We conclude that

$$f_S(W', a, x_i) = \frac{1}{m} \sum_{j=1}^{2m} a_j \langle x, w'_j \rangle \mathbf{1}[\langle x, w'_j \rangle > 0]$$

$$= \frac{1}{m} \sum_{j=1}^{m} a_i \langle x, w'_j \rangle \mathbf{1}[\langle x, w'_j \rangle > 0] + a_{j+m} \langle x, w'_{j+m} \rangle \mathbf{1}[\langle x, w'_{j+m} \rangle > 0]$$

$$= \frac{1}{m} \sum_{j=1}^{m} \langle x, w_j \rangle \mathbf{1}[\langle x, v_j \rangle > 0] = f(V, W, x_i). \qquad \square$$

We set $S_0 := S_0(X) = \{C(U) \mid U \subseteq [n], C(U) \neq \emptyset\}$. The following lemmas show that if our dataset is finite, then for every (convex) two-layer ReLU network there exists a (convex) two-layer ReLU of width at most $|S_0|$ such that their classification is the same for all $x_i, i \in [n]$. We note that $|S_0| \leq 2^n$, but we will show $O(n^{d-1})$ bounds on $|S_0|$ below.

**Lemma B.2.** *For any convex two-layer ReLU network $(V, W) \in \mathbb{R}^{m \times d} \times \mathbb{R}^{m \times d}$ let $S_V = \{C \in S_0 \mid \exists j \in [m] : v_j \in C\}$ and $m' = |S_V| \leq \min\{m, |S_0|\}$. Then there exists a convex two-layer ReLU network $(V', W') \in \mathbb{R}^{m' \times d} \times \mathbb{R}^{m' \times d}$ together with weights $\rho_1, \ldots, \rho_{m'}$ such that for all $i \in [n]$ it holds that*

$$f(V, W, x_i) = \frac{1}{m} \sum_{j=1}^{m} \langle x_i, w_j \rangle \mathbf{1}[\langle x_i, v_j \rangle > 0]$$

$$= \sum_{j=1}^{m'} \rho_j \langle x_i, w'_j \rangle \mathbf{1}[\langle x_i, v'_j \rangle > 0] = f(V', W', \rho, x_i).$$

Before proving the lemma we note that in this statement the weights are not necessary as we can replace $w'_j$ by $w'_j \cdot \rho_j$, however if we take the derivative with respect to $w_j$ then using the weighted version, we simplify the argument that the gradient also stays the same.

*Proof of Lemma B.2.* For each $C \in S_V$ let $J = \{j \in [m] \mid v_j \in C\}$ be the set of indices of orientation vectors in $C$. We set $v_C$ to be an arbitrary vector in $C$. We set $\rho_C = \rho_C(V) = |J|/m$ to be the fraction of orientation vectors in $C$. Further we set $w_C = \sum_{j \in J} w_j/|J|$. Then we have that

$$\sum_{C \in S_V} \rho_C \langle x_i, w_C \rangle \mathbf{1}[\langle x_i, v_C \rangle > 0] = \sum_{C \in S_V, i \in U(C)} \rho_C \langle x_i, w_C \rangle$$

$$= \frac{1}{m} \sum_{j \in [m], \langle v_j, x_i \rangle > 0} \langle x_i, w_j \rangle = f(V, W, x_i).$$

$\square$

There is an equivalent result for the usual two-layer networks:

**Lemma B.3.** *For any two-layer ReLU network $(W, a) \in \mathbb{R}^{m \times d} \times \{-1, 1\}^m$ let $S_V = \{(C, a_0) \in S_0 \times \{-1, 1\} \mid \exists j \in [m] : w_j \in C\}$ and $m' = |S_V| \leq \min\{m, 2|S_0|\}$. Then there exists a two-layer ReLU network $(W', a') \in \mathbb{R}^{m' \times d} \times \mathbb{R}^{m' \times d}$ together with weights $\rho_1, \ldots, \rho_{m'}$ such that for all $x_i, i \in [n]$ it holds that*

$$f_S(W, a, x_i) = \frac{1}{m} \sum_{j=1}^{m} a_j \langle x_i, w_j \rangle \mathbf{1}[\langle x_i, w_j \rangle > 0]$$

$$= \sum_{j=1}^{m'} \rho_j \langle x_i, w'_j \rangle \mathbf{1}[\langle x_i, w'_j \rangle > 0] = f_S(W', a', \rho, x_i).$$

*Proof.* Given a cone $C$ and a sign $a_0$ we set $J_{C,a_0} = \{j \in [m] \mid w_j \in C \text{ and } a_j = a_0\}$, $\rho_{C,a_0} = \rho_C(V) = |J_{C,a_0}|/m$. Further we set $w_{C,a_0} = \sum_{j \in J_{C,a_0}} w_j/|J_{C,a_0}|$. Note that $w_{C,a_0} \in C$. Then we have that

$$\sum_{(C,a_0) \in S_V} \rho_{C,a_0} a_0 \langle x_i, w_{C,a_0} \rangle \mathbf{1}[\langle x_i, w_{C,a_0} \rangle > 0] = \frac{1}{m} \sum_{j \in [m], a_j \langle w_j, x_i \rangle > 0} \langle x_i, w_j \rangle$$

$$= f(V, W, x_i).$$

$\square$

The following lemma shows that if there exists a cone $C(U)$ that is contained in a hyperplane $h = K(\{x\}) = K(\{-x\})$ for some $x \in \mathbb{R}^d$ then there exists a set $U' \subseteq X \cap K(C(U))$ such that both $x$ and $-x$ are an affine combination of vectors in $U'$. If $x \neq 0$ this implies that $U'$ is linearly dependent. We also note that if there exists a non-empty cone $C(U)$ with a Gaussian measure of $0$ then $C(U)$ is contained in a hyperplane.

**Lemma B.4.** *Let $U \subseteq [n]$ such that $C(U) \neq \emptyset$ and $x \in \mathbb{R}^d$ with $C(U) \subseteq K(\{x\})$. Then there exists $U_1 \subseteq X \cap K(C(U))$ such that for*

$$C'(U_1) = \{z \in \mathbb{R}^d \mid z = \sum_{i \in U_1} \alpha_i x_i \text{ for some } \alpha \in \mathbb{R}_{\geq 0}^{|U_1|}\}$$

*it holds that $x \in C'(U_1)$.*

*Proof.* We construct $U_1$ as follows: we start with $U_1 = \emptyset$ and then add points from $X \cap K(C(U))$ iteratively until $x \in C'(U_1)$. Since $C(U) \neq \emptyset$ there exists $z_U \in C(U)$. We further choose $z_U$ to be in the interior of $C(U)$, i.e., for all $i \in [n]$ we have that $\langle z_U, x_i \rangle = 0$ if and only if for all $z \in C(U)$ it holds that $\langle z, x_i \rangle = 0$.

If $x \notin C'(U_1)$ then we claim that we can find a point $z \in \mathbb{R}^d$ such that $\langle z, x \rangle \neq 0$ and for all $x' \in U_1$ we have that $\langle x', z \rangle \leq 0$: if $U_1 = \emptyset$ then we can set $z = x$. Otherwise let $x_0 \in C'(U_1) \cap \mathcal{S}^{d-1}$ minimize the distance between $x$ and $x_0$. Then we claim that $z = x - x_0$ satisfies $\langle z, x \rangle \neq 0$ and for all $x' \in U_1$ we have that $\langle x', z \rangle < 0$. Note that

$$\langle z, x \rangle = \langle x - x_0, x \rangle = 1 - \cos(\alpha) \neq 0$$

where $\alpha$ is the angle between $x$ and $x_0$. Further if there was a point $x' \in U_1$ with $\langle x', z \rangle > 0$ then there would be another point $x'' \in C'(U_1) \cap \mathcal{S}^{d-1}$ with $x'' = \frac{x_0 + \beta x'}{\|x_0 + \beta x'\|_2}$ for a sufficiently small $\beta \in \mathbb{R}_{>0}$ such that $x''$ is closer to $x$.

Since $\langle z_U, x \rangle = 0$, for any $\beta \in \mathbb{R}_{>0}$ it holds that $\langle z_U + \beta z, x \rangle = \langle \beta z, x \rangle \neq 0$ and $C(U) \subseteq K(\{x\})$. Thus, there must be a point $x_i$ such that $\mathbf{1}[\langle x_i, z_U \rangle > 0] \neq \mathbf{1}[\langle x_i, z_U + \beta z \rangle > 0]$. By choice of $z_U$ this implies that $\langle z_U, x_i \rangle = 0$ and $\langle \beta z, x_i \rangle = \langle z_U + \beta z, x_i \rangle > 0$ which in particular implies that $x_i \notin U_1$. Further since $\langle z_U, x_i \rangle = 0$ we also have that $x_i \in K(C(U))$ by choice of $z_U$. Thus we can add $x_i$ to $U_1$ and after iterating the previous steps at most $n$ times it holds that $x \in C'(U_1)$. $\square$

Next, we show that $|S_0|$ is actually bounded by $O(n^{d-1})$ which we can combine with the previous results to show that for any two-layer ReLU-network there exists a similar one whose width is bounded by at most $O(n^{d-1})$. The lemma follows as a direct consequence of Theorem 1 in (Cover, 1965). We prove the result for completeness.

**Lemma B.5.** *For any dataset $X$ it holds that $|S_0(X)| \leq 4n^{d-1}$. Further if $X$ is in general position and $n \geq d > 2$, i.e., any subset of $d$ points is linearly independent, then $|S_0(X) \setminus \{0\}| = \sum_{k=0}^{d-1} \binom{n}{k}$.*

*Proof.* Assume that $X$ is in general position and $n \geq d \geq 3$. Then all cones in $S_0$ have a Gaussian measure greater than 0, which in particular implies that two cones are separated by a face. Consider a connected non-empty subset $B \subseteq \mathcal{S}^{d-1}$ of the sphere such that $B = \bigcup_{C \in S_1} C$ where $S_1 \subseteq S_0$. We claim that the number of cones in $B$ equals the number of faces of the cones that are in $B$ plus 1 (we say that a face is in $B$ if it passes through the interior of $B$, i.e., excluding its boundaries). We show this via induction on the number of faces in $B$. The statement holds trivially if there exists no face in $B$. If there are more faces, we split $B$ along a hyperplane $h$ (a $(d-1)$-dimensional face) into subsets $B_1$ and $B_2$. We now apply the induction hypothesis which yields that the number of cones in $B_1$ equals the number of faces in $B_1$ plus 1 and the number of cones in $B_2$ equals the number of faces in $B_2$ plus 1. The number of faces on $B$ is exactly the number of faces in $B_1$ plus the number of faces in $B_2$ plus 1. To see this, note that any face that is completely contained in one of the $B_i$ remains a face in $B$ and if a face crosses $h$ then this creates a new face. Now we have an additional term of plus 2, but we also have one additional face corresponding to the splitting hyperplane $h$.

It remains to count the number of faces of $\mathcal{S}^{d-1}$ with respect to the set of hyperplanes $\{h_i \mid i \in [n]\}$ where $h_i = \{x \in \mathcal{S}^{d-1} \mid \langle x, x_i \rangle = 0\}$. Since $X$ is in general position, every subset $S \subseteq [n]$ of size at most $d-1$ represents a (non-trivial) face given by $\bigcap_{i \in S} h_i$. Further since $d > 2$, the intersection of any face with the sphere is connected. Thus, the number of cones that have a non-trivial intersection with $\mathcal{S}^{d-1}$ is equal to $\sum_{k=0}^{d-1} \binom{n}{k}$.

Finally, by combining all arguments, it holds for any dataset $X$ that

$$|S_0(X)| \leq 1 + \sum_{k=1}^{d-1} \binom{n}{k} \leq 2n^{d-1}.$$

If $X$ is not in general position, most of the arguments still apply, but some faces can be cones themselves. For instance, if there exist $x_i$ and $x_j$ such that $x_i = -x_j$, then there are cones that are completely contained in $K(x_i)$. In this case, faces can still divide one cone into two cones, but they can also be a cone themselves. Thus we still have that

$$|S_0(X)| \leq 1 + 2\sum_{k=1}^{d-1} \binom{n}{k} \leq 4n^{d-1}. \qquad \square$$

## C  SEPARATION MARGIN AND THE SMALLEST EIGENVALUE OF THE NTK

In this section, we consider two parameters used to bound the width for binary classification with logistic loss and regression with squared loss. We first define the parameter $\gamma$ which was introduced and analyzed in (Ji and Telgarsky, 2020; Munteanu et al., 2022) and $\lambda$ introduced in (Du et al., 2019c) and further analyzed in (Du et al., 2019a; Song and Yang, 2019; van den Brand et al., 2020; Munteanu et al., 2022) among others. We note that $\lambda$ is tightly related to separation and collinearity conditions studied earlier and extended, e.g. in (Li and Liang, 2018; Oymak and Soltanolkotabi, 2020).

## C.1 NTK SEPARATION MARGIN

Intuitively, $\gamma$ determines the separation margin of the NTK. Let $B = B^d = \{x \in \mathbb{R}^d \mid \|x\|_2 \leq 1\}$ be the unit ball in $d$ dimensions. We set $\mathcal{F}_B$ to be the set of functions $f$ mapping from $\text{dom}(f) = \mathbb{R}^d$ to $\text{range}(f) = B$. Let $\mu_{\mathcal{N}}$ denote the Gaussian measure on $\mathbb{R}^d$, specified by the Gaussian density with respect to the Lebesgue measure on $\mathbb{R}^d$.

**Definition C.1.** Given a data set $(X, Y) \in \mathbb{R}^{n \times d} \times \mathbb{R}^n$ and a map $\overline{v} \in \mathcal{F}_B$ we set

$$\gamma_{\overline{v}} = \gamma_{\overline{v}}(X, Y) := \min_{i \in [n]} y_i \int \langle \overline{v}(z), x_i \rangle \mathbf{1}[\langle x_i, z \rangle > 0] \, d\mu_{\mathcal{N}}(z).$$

We say that $\overline{v}$ is optimal if $\gamma_{\overline{v}} = \gamma(X, Y) := \max_{\overline{v}' \in \mathcal{F}_B} \gamma_{\overline{v}'}$.

We note that $\max_{\overline{v}' \in \mathcal{F}_B} \gamma_{\overline{v}'}$ always exists since $\mathcal{F}_B$ is a set of bounded functions on a compact subset of $\mathbb{R}^d$.

In (Munteanu et al., 2022) it was shown that for every map $\overline{v}$ there exists another map $\overline{v}'$ with $\gamma_{\overline{v}'} = \gamma_{\overline{v}}$ such that $\gamma_{\overline{v}'}$ is constant on all cones in $S_0$. In particular this implies that there exists a finite (weighted) convex neural network that satisfies

$$f(V, W, x_i) \geq \gamma_{\overline{v}'},$$

see also Theorem C.2. The network is given by $m = |S_0|$, $V$ consists of one representative $v_C \in C$ for each cone $C \in S_0$, $w_C = \overline{v}'(v_C)$ and $\rho_C = P(z \in C)$

Given $V$ we set $\mathcal{W} = \{W \in \mathbb{R}^{m \times d} \mid \|w_j\|_2 \leq 1 \text{ for all } j \in [m]\}$

$$\gamma_V = \max_{W \in \mathcal{W}} \min_{i \in [n]} f(V, W, x_i, \rho). \tag{6}$$

Moreover we set $W_V \in \mathcal{W}$ to be a weight matrix that attains the maximum, i.e., $\gamma_V = \min_{i \in [n]} f(V, W_V, x_i)$.

Given $V$, we further define the map $\overline{v}_{V,W} \in \mathcal{F}_B$ as follows: let $x \in \mathbb{R}^d$ then we set $\overline{v}_V(x) = \overline{v}_V(C(x)) = w_{C(x)}$ where $w_C = \sum_{j \in \{j \in [m] \mid v_j \in C\}} w_j / \rho_C$ as in Lemma B.3 if $\rho_C > 0$ and $\overline{v}_V(x) = 0$ otherwise.

## C.2 SMALLEST EIGENVALUE OF THE NTK KERNEL MATRIX

The kernel matrix $H \in \mathbb{R}^{n \times n}$ is defined by

$$H_{ij} = \mathbb{E}_{w \sim \mathcal{N}(0,I)}[\langle x_i, x_j \rangle \mathbf{1}[\langle x_i, w \rangle > 0, \langle x_j, w \rangle > 0]]$$

We set $\lambda = \lambda(X) = \lambda(H)$ to be the minimum eigenvalue of $H$. Given $V, \rho$ we further define the finite counterpart. To this end, recall that the default is $\rho_k = 1/m$ for all $k \in [m]$, which corresponds to previous work.

$$H_{ij}^{\text{dis}} = H_{ij}^{\text{dis}}(V) = \sum_{k=1}^{m} \rho_k \langle x_i, x_j \rangle \mathbf{1}[\langle x_i, v_k \rangle > 0, \langle x_j, v_k \rangle > 0]$$

and $\lambda_V = \lambda(X, V) = \lambda(H^{\text{dis}})$ to be the minimum eigenvalue of $H^{\text{dis}}$.

## C.3 CONVEX TWO-LAYER NETWORKS CAN ATTAIN THE INFINITE WIDTH LIMIT PARAMETERS

To show that a two-layer ReLU network converges, one usually shows that if the width $m$ of the network is large enough, then the finite width parameters $\gamma_W$ resp. $\lambda_W$ for the initial weight vectors $W$ are close to their infinite width limits $\gamma$ resp. $\lambda$. Further one shows that this does not change significantly during optimization. We note that for convex two-layer ReLU networks the values of $\gamma_V$ resp. $\lambda_V$ are determined at initialization since they depend only on $V$ and the data, which do not change during optimization. The parameters thus do not change at all which makes the second argument obsolete and simplifies the convergence analysis. The following theorem establishes that there exists a weighted network $(V, \rho)$ such that $\gamma_V = \gamma$ and $\lambda_V = \lambda$.

**Theorem C.2.** *Let $m = |S_0|$ and for $C \in S_0$ set $\rho_C = P(C)$ where $P(C)$ is the probability that a random vector is in $C$. Further let $v_C$ be any vector in $C$, and let $V$ be the matrix whose rows are the collection of all $v_C, C \in S_0$. Then it holds that $\lambda_V = \lambda$ and $\gamma_V = \gamma$.*

*Proof.* We recall that $\mathbb{R}^d = \dot{\bigcup}_{C \in S_0} C$ is the disjoint union of the cones as each point of $x \in \mathbb{R}^d$ belongs to a unique cone. To see the equivalence of the eigenvalues (in particular the smallest eigenvalues) observe that

$$H_{ij} = \mathbb{E}[\langle x_i, x_j \rangle \mathbf{1}[\langle x_i, w \rangle > 0, \langle x_j, w \rangle > 0]]$$

$$= \int \langle x_i, x_j \rangle \mathbf{1}[\langle x_i, w \rangle > 0, \langle x_j, w \rangle > 0] \, d\mu_{\mathcal{N}}(w)$$

$$= \sum_{C \in S_0} \int \langle x_i, x_j \rangle \mathbf{1}[\langle x_i, w \rangle > 0, \langle x_j, w \rangle > 0] \mathbf{1}[w \in C] \, d\mu_{\mathcal{N}}(w)$$

$$= \sum_{C \in S_0} P(C) \langle x_i, x_j \rangle \mathbf{1}[\langle x_i, v_C \rangle > 0, \langle x_j, v_C \rangle > 0]$$

$$= H_{ij}^{\mathrm{dis}}$$

using that by definition of the cones, the activation indicators $\mathbf{1}[\langle x_i, w \rangle > 0]$ and $\mathbf{1}[\langle x_j, w \rangle > 0]$ are constant for any cone $C$ if we restrict to $w \in C$. This implies that $\lambda_V = \lambda$.

By (Munteanu et al., 2022, Lemma $C.2$) there exists a map $\overline{v} \in \mathcal{F}_B$ that is constant on cones and such that

$$\gamma = \min_{i \in [n]} y_i \int \langle \overline{v}(z), x_i \rangle \mathbf{1}[\langle x_i, z \rangle > 0] \, d\mu_{\mathcal{N}}(z).$$

Similarly to the above, we also have for any $i \in [n]$ that

$$y_i \int \langle \overline{v}(z), x_i \rangle \mathbf{1}[\langle x_i, z \rangle > 0] \, d\mu_{\mathcal{N}}(z) = y_i \sum_{C \in S_0} P(C) \langle \overline{v}(v_C), x_i \rangle \mathbf{1}[\langle x_i, v_C \rangle > 0]$$

$$= y_i f(V, \overline{v}(V), \rho, x_i),$$

thus implying $\gamma_V = \gamma$. $\qquad\square$

# D  GRADIENT DESCENT WEIGHT UPDATES

In this section we study the loss function and its directional derivatives showing more similarities between the two variants of ReLU networks. Further, we show that the gradient behaves similarly in both formulations of the networks introduced in previous sections.

For a weighted convex neural network $(V, W, \rho)$ we set $L'(W)$ to be the gradient of the loss function $L(W)$, i.e.

$$L'(V, W, \rho)_j = \frac{\partial L(W)}{\partial w_j} = \sum_{i \in [n], \langle v_j, x_i \rangle > 0} \partial_{w_j} \ell(y_i, f(V, W, \rho, x_i))$$

We note that

$$L'(V, W, \rho) = \sum_{i=1}^{n} \ell'(y_i, f(V, W, \rho, x_i)) \nabla f(V, W, \rho, x_i)$$

and

$$(\nabla f(V, W, \rho, x_i))_j = \frac{\partial f(V, W, \rho, x_i)}{\partial w_j} = \rho_j \mathbf{1}[\langle x_i, v_j \rangle > 0] x_i.$$

Thus we also have that for any $W, W' \in \mathbb{R}^{m \times d}$

$$\langle \nabla f(V, W, \rho, x_i), W' \rangle = \sum_{j \in [m]} \langle \nabla f(V, W, \rho, x_i))_j, w'_j \rangle = f(V, W', \rho, x_i)$$

Similarly we have for a weighted neural network $(W, a, \rho)$ that

$$L'_S(W, a, \rho)_j = \frac{\partial L_S(W)}{\partial w_j} = \sum_{i \in [n], \langle w_j, x_i \rangle > 0} \partial_{w_j} \ell(y_i, f(W, a, \rho, x_i)),$$

$$L'_S(W, a, \rho) = \sum_{i=1}^{n} \ell'(y_i, f(W, a, \rho, x_i)) \nabla f(W, a, \rho, x_i).$$

and

$$(\nabla f(W, a, \rho, x_i))_j = \frac{\partial f(W, a, \rho, x_i)}{\partial w_j} = \rho_j a_j \mathbf{1}[\langle x_i, w_j \rangle > 0] x_i.$$

Recall that the default value for $\rho$ is the vector where all entries are equal to $1/m$.

In the following we consider the equivalent networks from previous sections. Intuitively, we start with a standard two-layer ReLU network $(W, a)$. We then define matrix transformations $T_1$, which maps $(W, a)$ to the equivalent network $(V, W')$ from Theorem B.1, $T_2$, which maps $(W, a)$ to the network $(V', W'', \rho)$ which is similar to $(V, W')$ from Lemma B.2, and $T_3$, which maps $(W, a)$ to the network $(W', a', \rho')$ from Lemma B.3.

Before we define the transformations more formally, we need some details about cones. Let $W \in \mathbb{R}^{m \times d}$ be any weight matrix and $a \in \{-1, 1\}^m$ to be a sign vector. For any vector $v \in \mathbb{R}^d$ we set $C(v)$ to be the cone $C \in S_0(X)$ containing $v$. We assume without loss of generality that there exists $m' \leq m$ such that for any distinct $j, j' \leq m'$ we have that $C(w_j) \neq C(w_{j'})$ and for any $j \in [m]$ there exists a unique index $i(j) \leq m'$ such that $C(w_j) = C(w_{i(j)})$. Further we assume without loss of generality that there exists $m'' \leq m$ such that for any distinct $j, j' \leq m''$ we have that $C(w_j) \neq C(w_{j'})$ or $a_j \neq a_{j'}$ and for any $j \in [m]$ there exists a unique index $i'(j) \leq m''$ such that $C(w_j) = C(w_{i'(j)})$ and $a_j = a_{i'(j)}$.

We now define the transformations more formally. We set $V = W$ and for $w \in \mathbb{R}^d$ we set $T_{1,j}(w) = a_j w$ and $T_1(W, a)$ to be the matrix whose $j$-th row is $T_{1,j}(w_j)$. Further we set $V' \in \mathbb{R}^{m' \times d}$ to be the matrix $V$ restricted to the first $m'$ rows and $\rho_j = |i^{-1}(j)|/m$ where $i^{-1}(j) = \{j' \in [m] \mid i(j') = j\}$. For $W' \in \mathbb{R}^{m \times d}$ and for $j \in [m']$ we set $T_2(W, a)$ to be the matrix with $j$-th row $T_{2,j}(W) = \sum_{j' \in i^{-1}(j)} w_{j'}/|i^{-1}(j)|$. Finally, we set $a' \in \{-1, 1\}^{m''}$ to be the vector with $a'_j = a_j$ and for $j \in [m'']$ we set $T_{3,j}(W) = \sum_{j' \in i'^{-1}(j)} w_{j'}/|i'^{-1}(j)|$ and $T_3(W, a)$ to be the matrix with rows $T_{3,j}(W)$ and weights $\rho'_j = |i'^{-1}(j)|/m$.

Then we get the following lemma:

**Lemma D.1.** *For the gradient it holds that*

$$L'(V, T_1(W, a)) = T_1(L'(V, W), a) \tag{7}$$

$$L'(V', T_2(W, a), \rho) = T_2(L'(V, W), a) \tag{8}$$

$$L'_S(T_3(W, a), a', \rho') = T_3(L'(V', W, \rho'), a). \tag{9}$$

*Further for any weight update $\Delta W$ such that for all $i \in [n]$ it holds that $\mathbf{1}[\langle x_i, w_j \rangle > 0] = \mathbf{1}[\langle x_i, w_j + \Delta w_j \rangle > 0]$ we have that*

$$L_S(W + \Delta W, a) = L(V, T_1(W + \Delta W, a))$$
$$= L(V', T_2(W + \Delta W, a), \rho) = L_S(T_3(W + \Delta W, a), a', \rho'). \tag{10}$$

*Proof.* In the following we use that for any $j \in [m]$ we have that

$$\mathbf{1}[\langle x_i, w_j \rangle > 0] = \mathbf{1}[\langle x_i, v_j \rangle > 0] = \mathbf{1}[\langle x_i, v'_{i(j)} \rangle > 0] = \mathbf{1}[\langle x_i, w_{i'(j)} \rangle > 0]$$

as well as $a_j = a_{i'(j)}$. We have that

$$T_1(L'(V, W), a)_j = a_j L'(V, W)_j$$

$$= \sum_{i=1}^{n} a_j \ell'(y_i, f(V, W, \rho, x_i))(\nabla f(V, W, \rho, x_i))_j = L'(V, T_1(W, a))_j.$$

Similarity we have that

$$T_2(L'(V,W),a)_j = \rho_j \sum_{j' \in i'^{-1}(j)} L'_S(V,W)_{j'}/|i^{-1}(j)|$$

$$= \rho_j \sum_{j' \in i'^{-1}(j)} \frac{1}{|i^{-1}(j)|} \sum_{i \in [n], \langle w_{j'}, x_i \rangle > 0} \ell'(y_i, f(V,W,\rho,x_i))a_{j'}x_i$$

$$= \rho_j \sum_{i \in [n], \langle w_j, x_i \rangle > 0} \ell'(y_i, f(V,W,\rho,x_i))x_i a_j = L'(V', T_2(W,a),\rho)$$

and

$$T_3(L'(V',W,\rho'),a)_j = \rho'_j \sum_{j' \in i^{-1}(j)} L'(V,W)_{j'}/|i^{-1}(j)|$$

$$= \rho_j \sum_{j' \in i^{-1}(j)} \frac{1}{|i^{-1}(j)|} \sum_{i \in [n], \langle w_{j'}, x_i \rangle > 0} \ell'(y_i, f(V,W,\rho',x_i))x_i$$

$$= \rho_j \sum_{i \in [n], \langle w_j, x_i \rangle > 0} \ell'(y_i, f(V,W,\rho',x_i))x_i = L'_S(T_3(W,a),a',\rho').$$

For the second part of the lemma, note that since $V = W$, we have

$$f(V,T_1(W+\Delta W,a),x_i) = \frac{1}{m}\sum_{j=1}^m \langle x_i, w_j + \Delta w_j \rangle \mathbf{1}[\langle x_i, v_j \rangle > 0]$$

$$= \frac{1}{m}\sum_{j=1}^m \langle x_i, w_j + \Delta w_j \rangle \mathbf{1}[\langle x_i, w_j \rangle > 0]$$

$$= \frac{1}{m}\sum_{j=1}^m \langle x_i, w_j + \Delta w_j \rangle \mathbf{1}[\langle x_i, w_j + \Delta w_j \rangle > 0]$$

$$= f(W+\Delta W, T_1(W+\Delta W,a),x_i).$$

and $f(W+\Delta W, T_1(W+\Delta W,a),x_i) = f_S(W+\Delta W,a,x_i)$ by Lemma B.1 and thus also $L_S(W+\Delta W,a) = L(V,T_1(W+\Delta W,a))$. The equations $L(V,T_1(W+\Delta W,a)) = L(V',T_2(W+\Delta W,a),\rho)$ and $L_S(W+\Delta W,a) = L_S(T_3(W+\Delta W,a),a',\rho')$ follow similarly.

$\square$

## E  GRADIENT DESCENT APPROXIMATION RESULTS

Next we establish an approximation result for convex two-layer ReLU networks with logistic loss, i.e. $\ell(r) = \ln(1 + \exp(-r))$.

Recall that we initialize $W = W_0$ to be a zero matrix and apply gradient descent to update the weights for $t \geq 0$ in an iterative manner

$$W_{t+1} = W_t - \eta_t L'(W_t)$$

where $\eta_t \in \mathbb{R}_{\geq 0}$ is a learning rate parameter and $L'(W_t)$ is the gradient of the loss function $L(W_t)$ at $W_t$

$$L'(V,W,\rho) = \sum_{i=1}^n \ell'(y_i, f(V,W,\rho,x_i))\nabla f(V,W,\rho,x_i)$$

We note that $-\ell'(r) \leq \min\{1, \ell(r)\}$ which in particular implies that $L(W)$ is a $\frac{n}{m}$-Lipschitz function, which becomes $\frac{L(W)}{m}$-Lipschitz if we restrict to a small radius around $W$. Combining these properties of the convex loss function $L$ and the fact that $\max_{j \in [m]} \|w_j - w'_j\|_2 \leq 2B$ for any $W, W' \in \mathcal{W}_B = \{W \in \mathbb{R}^{m \times d} \mid \max_{j \in [m]} \|w_j\|_2 \leq B\}$ implies a similar bound in Frobenius norm canceling the factor $m$ and yields that gradient descent converges to within a factor 2 to the optimal solution $W^*$ using *standard* gradient descent analyses (Nesterov, 2004; Bubeck, 2015) in roughly $B^2$ iterations. The following lemma guarantees that there exists a real number $B \in \mathbb{R}_{>0}$ that is not too large and a near-optimal solution within the restricted domain $W \in \mathcal{W}_B$ such that $\min_{i \in [n]} y_i f(V,W,x_i)$ is close to $\gamma$.

**Lemma E.1.** *Let $\gamma_V > 0$ as defined in Equation* (6). *Let $\ell(r) = \ln(1 + \exp(-r))$ and $\varepsilon > 0$. Let $B \geq (\ln(4) + \ln(n))/(\varepsilon\gamma_V)$. Let $W^* \in \mathcal{W}_B$ minimize $L(W)$ and let $W \in \mathcal{W}_B$ be any solution such that $L(W) \leq 2L(W^*)$. Then it holds that $\gamma_V \geq \min_{i \in [n]} y_i f(V, W/B, x_i) \geq (1 - \varepsilon)\gamma_V$*

*Proof.* First recall that $W_V \in \mathcal{W} = \{W \in \mathbb{R}^{m \times d} \mid \|w_j\|_2 \leq 1 \text{ for all } j \in [m]\}$ was defined to be a maximizer of $\max_{W \in \mathcal{W}} \min_{i \in [n]} y_i f(V, W, x_i) = \gamma_V$, see Equation (6).

We set $\overline{W} = BW_V$ and note that $\overline{W} \in \mathcal{W}_B$. Using that $\exp(-r)/(1 + \exp(-r)) = -\ell'(r) \leq \ell(r) \leq \exp(-r)$ and $\ln(4) + \ln(n) - \gamma_V B \leq 0$ we get that

$$
\begin{aligned}
L(W^*) \leq L(\overline{W}) &= \sum_{i=1}^{n} \ell(y_i f(V, BW_V, x_i)) \\
&\leq \sum_{i=1}^{n} \ell(\gamma_V B) \\
&= n\,\ell(\gamma_V B) \\
&\leq n \exp(-\gamma_V B) \\
&= \frac{1}{2} \cdot \frac{\exp(\ln(4) + \ln(n) - \gamma_V B)}{2} \\
&\leq \frac{1}{2} \cdot \frac{\exp(\ln(4) + \ln(n) - \gamma_V B)}{1 + \exp(\ln(4) + \ln(n) - \gamma_V B)} \\
&\leq \ell((1 - \varepsilon)\gamma_V B)/2
\end{aligned}
$$

Now let $W \in \mathcal{W}_B$ be any solution with $L(W) \leq 2L(W^*)$. Then for any $i \in [n]$ we have that

$$\ell(y_i f(V, W, x_i)) \leq L(W) \leq 2L(W^*) \leq \ell((1 - \varepsilon)\gamma_V B)$$

which by strict monotonicity of $\ell$ and linearity of $f$ implies that $y_i f(V, W/B, x_i) \geq (1 - \varepsilon)\gamma_V$. $\square$

Next we show that $\gamma$ and $\gamma_V$ can be related to each other. To prove this we will use the Hoeffding bound.

**Lemma E.2** (Hoeffding bound (Hoeffding, 1963)). *Let $X_1, \ldots, X_n$ denote $n$ independent bounded variables in $[a_i, b_i]$. Let $X = \sum_{i=1}^{n} X_i$. Then we have*

$$\Pr[|X - \mathbb{E}[X]| \geq t] \leq 2 \exp\left(-\frac{2t^2}{\sum_{i=1}^{n}(b_i - a_i)^2}\right).$$

**Theorem E.3.** *Assume $V \in \mathbb{R}^{m \times d}$ is initialized with i.i.d. Gaussians and $0 < \delta \leq \varepsilon$. For any $m$ it holds that $\mathbb{E}\gamma_V \leq \gamma$ over the Gaussian measure. Further, if the network width is $m \geq c \cdot (\varepsilon\delta\gamma)^{-2} \ln(n/\varepsilon)$ for an absolute constant $c > 0$, then with probability at least $1 - \delta$ it holds that $(1 + \varepsilon)\gamma \geq \gamma_V \geq (1 - \varepsilon)\gamma$.*

*Proof.* Given $V, V' \in \mathbb{R}^{m \times d}$ we define that $V \simeq V'$ if for all cones $C \in S_0$ it holds that $\rho_C(V) = \rho_C(V')$. We set $\mathcal{V}$ to be the set of equivalence classes with respect to $\simeq$ and given $\tilde{V} \in \mathcal{V}$ we set $P(\tilde{V})$ to be the probability that a randomly drawn set $V' \in \tilde{V}$, i.e., $V' \simeq V$. For any cone $C \in S_0$ let $\mathcal{V}_C = \{\tilde{V} \in \mathcal{V} \mid \exists j \in [m] : \tilde{v}_j \in C\}$ the set of orientation matrices such that there exists at least one orientation in $C$. Further we set $P(C)$ to be the probability that a random vector $z \in \mathbb{R}^d$ is in $C$ and $P(\tilde{V} \mid v_1' \in C)$ to be the probability that a randomly drawn $V'$ is equivalent to $V$ given that the first vector of $V'$ is in $C$. We partition each cone $C \in S_0$ into subregions $C = \dot{\bigcup}_{\tilde{V} \in \mathcal{V}_C} C(\tilde{V})$ such that the probability that a random vector $z \in C$ is in $C(\tilde{V})$ is $P(\tilde{V} \mid v_1' \in C)$. Note that this yields a partition $\mathbb{R}^d = \dot{\bigcup}_{C \in S_0, \tilde{V} \in \mathcal{V}_C} C(\tilde{V})$.

Using Bayes' theorem and the fact that $P(v_1 \in C \mid \tilde{V}) = \rho_C$ we get that

$$P(\tilde{V} \mid v_1 \in C)P(C) = P(v_1 \in C \mid \tilde{V})P(\tilde{V}) = P(\tilde{V})\rho_C.$$

For any $z \in C(\tilde{V})$ we set $\overline{v}(C) = \overline{v}(z) = \overline{v}_V(z) \in B$. Then for any $i \in [n]$ we have that

$$\gamma \geq \gamma_{\overline{v}} = y_i \int \langle \overline{v}(z), x_i \rangle \mathbf{1}[\langle x_i, z \rangle > 0] \, d\mu_{\mathcal{N}}(z)$$

$$= y_i \sum_{C \in S_0} P(C) \sum_{\tilde{V} \in \mathcal{V}_C} P(\tilde{V} \mid v_1 \in C) \langle \overline{v}_V(C), x_i \rangle \mathbf{1}[\langle x_i, v_C \rangle > 0]$$

$$= y_i \sum_{C \in S_0} \sum_{\tilde{V} \in \mathcal{V}_C} P(\tilde{V}) \rho_C \langle \overline{v}_V(C), x_i \rangle \mathbf{1}[\langle x_i, v_C \rangle > 0]$$

$$= \sum_{\tilde{V} \in \mathcal{V}} P(\tilde{V}) \sum_{j \in [m]} y_i \langle \overline{v}_V(v_j), x_i \rangle \mathbf{1}[\langle x_i, v_j \rangle > 0] = \mathbb{E}\gamma_V.$$

For the second part assume that $m \geq c \cdot (\varepsilon \delta \gamma)^{-2} \ln(n/\varepsilon)$ and let $\overline{v}$ be optimal, i.e. $\gamma = \gamma_{\overline{v}}$. We first fix $i \in [n]$ and set $Z_j = y_i \langle \overline{v}_V(v_j), x_i \rangle \mathbf{1}[\langle x_i, v_j \rangle > 0]$ and note that $Z_j \in [-1, 1]$ and $\mathbb{E}(Z_j) \geq \gamma$. Then using Hoeffding's bound for $Z = \sum_{j=1}^m Z_j$, we get that

$$\Pr[|Z - \mathbb{E}[Z]| \geq m\gamma\varepsilon] \leq 2 \exp\left(-\frac{2(m\gamma\varepsilon)^2}{4m}\right) \leq \varepsilon/n.$$

Using the union bound over all $i$ we have that $\gamma_V \geq (1 - \varepsilon)\gamma$ holds with probability at least $1 - \varepsilon$. We further have that $\gamma_V \in [0, 1]$ and

$$\gamma \geq \mathbb{E}\gamma_V \geq (1 - \varepsilon)(1 - \varepsilon)\gamma + P(\gamma_V \geq (1 + \varepsilon')\gamma)\varepsilon'\gamma$$

$$\geq (1 - 3\varepsilon)\gamma + P(\gamma_V \geq (1 + \varepsilon')\gamma)\varepsilon'\gamma$$

We conclude that

$$P(\gamma_V \geq (1 + \varepsilon')\gamma)\varepsilon' \leq 3\varepsilon.$$

Now choosing $\varepsilon' = 3\varepsilon/\delta$ gives us $P(\gamma_V \geq (1 + \varepsilon')\gamma) \leq \delta$. Thus the second part follows by substituting $\varepsilon$ by $\varepsilon'$. $\square$

# F  Two important data examples

## F.1  The alternating circle and why it might be hard to prove that a two-layer ReLU network of linear width suffices for arbitrarily small error

Consider the following set of $n$ points:
$x_k = \left(\cos\left(\frac{2k\pi}{n}\right), \sin\left(\frac{2k\pi}{n}\right)\right)$ and $y_k = (-1)^k$. The dataset consists of equidistant points on the circle with alternating labels. It has been used in (Munteanu et al., 2022) to derive lower bounds of different strengths on the width of two-layer ReLU networks. Since the labels are alternating this can be considered a hard dataset for two-layer ReLU networks and we will use it to show that if one wants to prove that a network of linear width suffices one will need more advanced proof techniques than the ones established previously.

(Munteanu et al., 2022) proved that for the specific choice of $\overline{v} \in \mathcal{F}_B$ used in the upper bounds of (Ji and Telgarsky, 2020), there exists an index $i \in [n]$ with

$$\frac{1}{m} \sum_{s=1}^m y_i \langle \overline{v}(w_s), x_i \rangle] \mathbf{1}\left[\langle x_i, w_s \rangle > 0\right] \leq 0.$$

with constant probability if the network has smaller width than $m < c \cdot \gamma^{-2}$. We will strengthen the lower bound of (Munteanu et al., 2022) by showing that this holds for any fixed $\overline{v} \in \mathcal{F}_B$. This strengthened lower bound leaves two possible options, one of which is true:

- $m = \Omega(\gamma^{-2})$ is indeed a general lower bound, i.e., lower $m$ precludes the existence of a separating $\overline{v}$,

- or showing $m = o(\gamma^{-2})$ requires to choose $\overline{v}$ *adaptively* to the size $m$ subsample of neurons.

In particular, our new bound thus shows that existing non-adaptive proof techniques are not sufficient to show $m = o(\gamma^{-2})$ upper bounds.

We will need the following lemma:

**Lemma F.1** ((StEx, 2011)). *For any $a, b \in \mathbb{R}$ and $\tilde{n} \in \mathbb{N}$ it holds that*

$$\sum_{k=0}^{\tilde{n}-1} \cos(a + kb) = \frac{\cos(a + (\tilde{n}-1)b/2)\sin(\tilde{n}b/2)}{\sin(b/2)}.$$

*Proof.* We use $\mathbf{i}$ to denote the imaginary unit defined by the property $\mathbf{i}^2 = -1$. From Euler's identity we know that $\cos(a + kb) = \text{Re}(e^{\mathbf{i}(a+kb)})$ and $\sin(a + kb) = \text{Im}(e^{\mathbf{i}(a+kb)})$. Then

$$\sum_{k=0}^{\tilde{n}-1} \cos(a + kb) = \sum_{k=0}^{\tilde{n}-1} \text{Re}\left(e^{\mathbf{i}(a+kb)}\right)$$

$$= \text{Re}\left(\sum_{k=0}^{\tilde{n}-1} e^{\mathbf{i}(a+kb)}\right)$$

$$= \text{Re}\left(e^{\mathbf{i}a} \sum_{k=0}^{\tilde{n}-1} (e^{\mathbf{i}b})^k\right)$$

$$= \text{Re}\left(e^{\mathbf{i}a} \frac{1 - e^{\mathbf{i}b\tilde{n}}}{1 - e^{\mathbf{i}b}}\right)$$

$$= \text{Re}\left(e^{\mathbf{i}a} \frac{e^{\mathbf{i}b\tilde{n}/2}(e^{-\mathbf{i}b\tilde{n}/2} - e^{\mathbf{i}b\tilde{n}/2})}{e^{\mathbf{i}b/2}(e^{-\mathbf{i}b/2} - e^{\mathbf{i}b/2})}\right)$$

$$= \frac{\cos(a + (\tilde{n}-1)b/2)\sin(\tilde{n}b/2)}{\sin(b/2)}.$$

$\square$

To keep the technical part simple, we will assume that $n \mod 4 = 0$. However, it is possible to get similar results for other $n$ as well. The next lemma allows us to show that for the given data example for any map $\overline{v} \in \mathcal{F}_B$ there exists an index $i$ such that the variance of the random variable defined by $Z_i = \langle \overline{v}(z), y_i x_i \rangle \mathbf{1}[\langle x_i, z \rangle > 0]$ is lower bounded by $\Omega(\gamma_X^{-2}\mathbb{E}(Z_i)^2)$.

**Lemma F.2.** *Let $X$ be the* alternating points on the circle *dataset defined above with $n \equiv 0 \mod 4$. Then the following holds for absolute constants $c$, and $c'$:*

- *The separation margin is given by $\gamma = \gamma_X = \Theta(1/n)$*

- *For any fixed map $\overline{v} \in \mathcal{F}_B$ it holds that*

$$\frac{\frac{1}{n} \cdot \sum_{i \in n} \int \langle \overline{v}(z), y_i x_i \rangle \mathbf{1}[\langle x_i, z \rangle > 0] \, d\mu_{\mathcal{N}}(z)}{\frac{1}{n} \cdot \sum_{i \in n} \int |\langle \overline{v}(z), y_i x_i \rangle| \mathbf{1}[\langle \overline{v}(z), y_i x_i \rangle < 0]\mathbf{1}[\langle x_i, z \rangle > 0] \, d\mu_{\mathcal{N}}(z)} \leq c/n$$

- *there exists at least one index $i \in [n]$ such that*

$$\frac{\int \langle \overline{v}(z), y_i x_i \rangle \mathbf{1}[\langle x_i, z \rangle > 0] \, d\mu_{\mathcal{N}}(z)}{\int |\langle \overline{v}(z), y_i x_i \rangle| \mathbf{1}[\langle \overline{v}(z), y_i x_i \rangle < 0]\mathbf{1}[\langle x_i, z \rangle > 0] \, d\mu_{\mathcal{N}}(z)} \leq c'\gamma_X$$

*and $\text{Var}(Z_i) \geq \Omega(\gamma_X^{-2}\mathbb{E}(Z_i)^2)$.*

*Proof.* Note that any cone $C = C(U) \in S_0$ is related to a set of points of the form $U = \{x_k, x_{k+1}, \ldots, x_{k+\lceil n/2 \rceil}\}$ or $U = \{x_k, x_{k+1}, \ldots, x_{k+\lceil n/2 \rceil - 1}\}$. Note that any cone with non-zero probability is related to a subset containing exactly half of the points thus we will restrict to those sets. By symmetry we can assume without loss of generality that $U = \{x_1, \ldots, x_{n/2}\}$.

Now let $z \in \mathbb{R}^2$ be any vector. In the following we calculate the contribution of $z$ as a weight vector. We rotate the circle so that $z = (\alpha, 0)$. Then by Lemma F.1 we have that

$$\sum_{k=1}^{n/2} y_k \langle x_k, z \rangle$$

$$= \sum_{i=1}^{n/2} (-1)^k \alpha \cos(r_0 + 2\pi \cdot k/n)$$

$$= \sum_{i=1}^{n/4} \alpha \cos(r_0 + k \cdot 4\pi/n) - \sum_{i=1}^{n/4} \alpha \cos(r_0 + 2\pi/n + k \cdot 4\pi/n)$$

$$= \alpha \cdot \left( \frac{\cos(r_0 + (n/4 - 1) \cdot \pi/n) \sin(\pi/2)}{\sin(2\pi/n)} - \frac{\cos(r_0 + 2\pi/n + (n/4 - 1) \cdot \pi/n) \sin(\pi/2)}{\sin(2\pi/n)} \right)$$

$$= \alpha \cdot \left( \frac{(\cos(r_0 + (n/4 - 1) \cdot \pi/n) - \cos(r_0 + 2\pi/n + (n/4 - 1) \cdot \pi/n))}{\sin(2\pi/n)} \right)$$

where $r_0$ is given by the rotation. Note that $\sin(2\pi/n) = \Theta(1/n)$. Further,

$$\cos(r_0 + (n/4 - 1) \cdot \pi/n) - \cos(r_0 + 2\pi/n + (n/4 - 1) \cdot \pi/n) = \int_{r_0 + (n/4-1) \cdot \pi/n}^{r_0 + 2\pi/n + (n/4-1) \cdot \pi/n} - \sin(t)\, dt$$

is maximized for $r_0 = (3/2)\pi$ in which case it is in $\Theta(1/n)$ as well. Choosing $\alpha = 1$, we thus get that

$$\sum_{i=1}^{n} \int \langle \overline{v}(z), y_i x_i \rangle \mathbf{1}[\langle x_i, z \rangle > 0]\, d\mu_{\mathcal{N}}(z) = O(1)$$

By averaging over $i \in [n]$, we get that $\gamma_X = O(1/n)$. Further using the argumentation above using the right $z$ for each cone and using the symmetry of the instance, we get that $\gamma_X = \Theta(1/n)$.

For the second part, note that for any $z \in \mathbb{R}^2$ it holds for one third of the indices $k \in [n]$ that the term $(-1)^k \alpha \cos(r_0 + 2\pi \cdot k/n)$ is negative and $\cos(r_0 + 2\pi \cdot k/n) \geq \cos(\pi/3) = 1/2$. Consequently by a similar argumentation as above we have that

$$\frac{\sum_{i \in n/2} \langle z, y_i x_i \rangle}{\sum_{i \in n/2} |\langle z, y_i x_i \rangle| \mathbf{1}[\langle z, y_i x_i \rangle < 0]\, d\mu_{\mathcal{N}}(z)} = \frac{O(1)}{n/12} = O(1/n)$$

and thus it also holds for any map $\overline{v} \in \mathcal{F}_B$ that

$$\frac{\frac{1}{n} \cdot \sum_{i \in n} \int \langle \overline{v}(z), y_i x_i \rangle \mathbf{1}[\langle x_i, z \rangle > 0]\, d\mu_{\mathcal{N}}(z)}{\frac{1}{n} \cdot \sum_{i \in n} \int |\langle \overline{v}(z), y_i x_i \rangle| \mathbf{1}[\langle \overline{v}(z), y_i x_i \rangle < 0] \mathbf{1}[\langle x_i, z \rangle > 0]\, d\mu_{\mathcal{N}}(z)} = O(1/n).$$

For the last part of the lemma, we observe by averaging that there exists one index $i \in [n]$ such that

$$\frac{\int \langle \overline{v}(z), y_i x_i \rangle \mathbf{1}[\langle x_i, z \rangle > 0]\, d\mu_{\mathcal{N}}(z)}{\int |\langle \overline{v}(z), y_i x_i \rangle| \mathbf{1}[\langle \overline{v}(z), y_i x_i \rangle < 0] \mathbf{1}[\langle x_i, z \rangle > 0]\, d\mu_{\mathcal{N}}(z)} = O(1/n).$$

The numerator is exactly the expected value of $Z_i$ and the denominator is a lower bound on the root of the variance of $Z_i$, where we use that $\mathbb{E}(Z_i)$ is non-neagtive

$$\mathrm{Var}(Z_i) = \int ((\mathbb{E}(Z_i) - \overline{v}(z), y_i x_i) \mathbf{1}[\langle x_i, z \rangle > 0])^2\, d\mu_{\mathcal{N}}(z)$$

$$\geq \int (|\langle \overline{v}(z), y_i x_i \rangle| \mathbf{1}[\langle \overline{v}(z), y_i x_i \rangle < 0] \mathbf{1}[\langle x_i, z \rangle > 0])^2\, d\mu_{\mathcal{N}}(z)$$

$$\geq \left( \int |\langle \overline{v}(z), y_i x_i \rangle| \mathbf{1}[\langle \overline{v}(z), y_i x_i \rangle < 0] \mathbf{1}[\langle x_i, z \rangle > 0]\, d\mu_{\mathcal{N}}(z) \right)^2.$$

Thus, it follows that $\mathrm{Var}(Z_i) \geq \Omega(\gamma_X^{-2} \mathbb{E}(Z_i)^2)$. $\qquad \square$

Using Lemma F.2 we prove for our *alternating points on the circle* dataset $X$ that if the width of the network is $m = o(\gamma^{-2})$ then with constant probability there exists an index $i \in [n]$ such that $\sum_{j \in [m]} \langle \overline{v}(z_j), y_i x_i \rangle \mathbf{1}[\langle x_i, z_j \rangle > 0] \leq 0$.

To show this we will use the following lemma:

**Lemma F.3** ((Feller, 1943)). *Let $Z$ be a sum of independent random variables, each attaining values in $[0, 1]$, and let $\sigma = \sqrt{\mathrm{Var}(Z)} \geq 200$. Then for all $t \in [0, \frac{\sigma^2}{100}]$ we have*

$$\Pr[Z \geq \mathbb{E}[Z] + t] \geq c \cdot \exp(-t^2/(3\sigma^2))$$

*where $c > 0$ is some fixed constant.*

**Theorem F.4.** *Let $X$ be the* alternating points on the circle *dataset with $n$ divisible by $4$ and let $W \in \mathbb{R}^{m \times 2}$ be a matrix consisting of $m$ Gaussian's. Then there is a constant $c_0 > 0$ such that if $m \leq c_0 \gamma_X^{-2}$, then for any $\overline{v} \in \mathcal{F}_B$ there exists an index $i \in [n]$ such that with constant probability*

$$\frac{1}{m} \sum_{s=1}^{m} y_i \langle \overline{v}(w_s), x_i \rangle] \mathbf{1}\left[\langle x_i, w_s \rangle > 0\right] \leq 0.$$

*Proof.* Let $\overline{v} \in \mathcal{F}_B$ and for $i \in [n]$ let $Z_i = \langle \overline{v}(z), y_i x_i \rangle \mathbf{1}[\langle x_i, z \rangle > 0]$. By Lemma F.2 there exists $i \in [n]$ with $\mathrm{Var}(Z_i) \geq \Omega(\gamma_X^{-2} \mathbb{E}(Z_i)^2)$. Note that $\frac{1}{m} \sum_{s=1}^{m} y_i \langle \overline{v}(w_s), x_i \rangle] \mathbf{1}\left[\langle x_i, w_s \rangle > 0\right]$ can be viewed as a random variable $Z$ which is a sum of independent random variables with the distribution of $Z_i$ divided by $m$. We further set $Z_i' = \frac{1 - Z_i}{2}$ and

$$Z' = \left( m - \sum_{s=1}^{m} y_i \langle \overline{v}(w_s), x_i \rangle] \mathbf{1}\left[\langle x_i, w_s \rangle > 0\right] \right) / 2$$

and note that $Z'$ is the sum of independent random variables with the distribution of $Z_i'$. Further, $\mathbb{E}(Z') = (m - m\mathbb{E}(Z_i))/2$,

$$\mathrm{Var}(Z') = m \cdot \mathrm{Var}(Z_i') = m \cdot \mathrm{Var}(Z_i)/4 \geq mc_0 \gamma_X^{-2} \mathbb{E}(Z_i)^2$$

and $Z' \geq m/2$ if and only if $Z < 0$. Thus, we can apply Lemma F.3 with $t = m\mathbb{E}(Z_i)/2$ to get that

$$\Pr[Z' \geq \mathbb{E}[Z'] + t] \geq c \cdot \exp(-t^2/(3\sigma^2)) \geq c \cdot \exp(-c_1),$$

for an absolute constant $c_1$, which finishes the proof. $\qquad\square$

## F.2 THE 3-DIMENSIONAL HYPERCUBE AND CONES OF MEASURE ZERO

The next example we want to consider is the 3-dimensional hypercube with parity labels. More precisely the dataset is given by $X = \{-1, 1\}^3$ and for $x \in X$ we set $y_x = x_1 x_2 x_3$, i.e., $y_x = 1$ if the number of 1's in $x$ is odd, otherwise $y_x = -1$.

This toy example is interesting for the following reason: we have that $\gamma_X = 0$ and we will show in the following that there exists no two-layer ReLU network that correctly classifies all points. However, there exists a convex two-layer ReLU network that classifies all points correctly using cones of measure $0$.

**Theorem F.5.** *Let $X$ be the 3-dimensional hypercube with parity labels. Then the following statements hold:*

- *$\gamma_X = 0$,*

- *there exists no two-layer ReLU network that classifies all points correctly,*

- *there exists a convex two-layer ReLU network that classifies all points correctly.*

*Proof.* The first item was proven in Lemma C.7 of (Munteanu et al., 2022) in general dimension including the special case with $d = 3$.

The second claim can be reformulated to state that for any weight vector $w \in \mathbb{R}^3$ it holds that

$$\sum_{i=1}^{8} y_i \langle x_i, w \rangle \mathbf{1} \left[ \langle x_i, w_s \rangle > 0 \right] = 0.$$

To see this, observe that for any $w$ there exists $i \in [8]$ that maximizes $\langle x_i, w \rangle$. Let $S \subseteq X$ consist of $x_i$ and the three Hamming neighbors of $x_i$. Then we have that

$$\sum_{i=1}^{8} y_i \langle x_i, w \rangle \mathbf{1} \left[ \langle x_i, w \rangle > 0 \right] = \sum_{x \in S} y_x \langle x, w \rangle = \left\langle \sum_{x \in S} y_x x, w \right\rangle = 0$$

since $\sum_{x \in S} y_x x = 0$.

Next observe that this implies that also for any $W \in \mathbb{R}^{m \times 3}$ and $a \in \{-1, 1\}^m$ it holds that

$$\sum_{i=1}^{8} \sum_{j=1}^{m} a_j \langle x_i, w_j \rangle \mathbf{1} \left[ \langle x_i, w_j \rangle > 0 \right] = \sum_{j=1}^{m} a_j \sum_{i=1}^{8} \langle x_i, w_j \rangle \mathbf{1} \left[ \langle x_i, w_j \rangle > 0 \right] = 0.$$

which means there exists at least one point $x \in X$ with $y_i f(W, a, x) \leq 0$.

For the last item it suffices to define a convex two-layer ReLU network that classifies all points correctly. To check and explain the idea we give a concrete description of the dataset:

$$x_1 = (1, 1, 1), x_2 = (1, 1, -1), x_3 = (1, -1, 1), x_4 = (-1, 1, 1),$$
$$x_5 = (1, -1, -1), x_6 = (-1, 1, -1), x_7 = (-1, -1, 1), x_8 = (-1, -1, -1)$$
$$y_1 = y_5 = y_6 = y_7 = 1, y_2 = y_3 = y_4 = y_8 = -1$$

We set $m = 8$ and

$$v_1 = (1, 1/2, 1/2), v_2 = (1, 1/2, -1/2), v_3 = (1, -1/2, 1/2), v_4 = (1, -1/2, -1/2),$$
$$v_5 = -(1, 1/2, 1/2), v_6 = (1, 1/2, -1/2), v_7 = -(1, -1/2, 1/2), v_8 = -(1, -1/2, -1/2)$$
$$w_1 = w_4 = w_6 = w_7 = -e_1, w_2 = w_3 = w_5 = w_8 = e_1,$$

where $e_1 = (1, 0, 0)$ is the first standard unit vector.

The idea of this network is that for each orientation vector $v_j$, there are exactly three points $x_i$ with $\langle x_i, v_j \rangle > 0$. Further for any point $x_i$ there are exactly two vectors $v_j$ with $\langle x_i, v_j \rangle = 1$ and $\langle y_i x_i, w_j \rangle = 1$, then there exists one vector $v_j$ with $\langle x_i, v_j \rangle = 2$ and $\langle y_i x_i, w_j \rangle = -1$ and for the remaining vectors $v_j$ we have that $\langle x_i, v_j \rangle \leq 0$ which implies that

$$y_i f(V, W, x_i) = 1$$

and thus all points are classified correctly. $\qquad\square$

As a remark, we note that if a convex network is initialized via random Gaussians as orientation vectors, then the network will not converge to a network that classifies all points correctly since there needs to be at least one orientation $v_j$ with $\langle x_i, v_j \rangle = 0$ for some $i$ but this event occurs with probability 0.

We additionally remark that the alternating circle with $n = 2 \mod 4$ and $n \geq 6$ has similar properties as the 3-dimensional hypercube with parity labels.

