# OpenReview forum: "NTK with Convex Two-Layer ReLU Networks"
_ICLR.cc/2026/Conference — Submitted to ICLR 2026_

### Official Review · Reviewer_cx59 · 2025-10-25

**Soundness:** 4
**Presentation:** 2
**Contribution:** 3
**Rating:** 4
**Confidence:** 3

**Summary:**

The authors use a formulation of shallow relu networks called ‘convex networks’ which decouple where neurons turn on and off from their actual numeric outputs. The activation region layout is thus fixed at initialization, and only the neuron outputs are trained. The convex network formulation makes it so that data points do not change activation regions during the course of training, so that the optimization problem becomes convex in parameter space. This nice property allows the authors to establish when infinite width networks could be replaced by finite width networks and still maintain the NTK separation margin.

**Strengths:**

1. The authors are extensively aware of the related literature
2. Use of the convex network formulation is a clever way to make results tractable, while also being interesting in its own right.
3. The authors present an impressive variety of results.

**Weaknesses:**

1. 43 up to a factor OF two
2. 148 ‘this motivates to’
3. 190 did you mean to say ‘at initialization’ rather than ‘at activation’?
4. 229 improve should be improves
5. 235 should say (omitting parameters other than gamma)
6. The paper does a lot of things, but that might work to its detriment, for example, the contributions section has 8 items in it, but not all of them feel like they’re the central point of the paper.
7. The paper is very dense and symbol heavy with no figures to break it up. Although the main content of the paper is mathematical, the example data problems at least could probably benefit from figures?
8.  Perhaps the paper could dedicate a little more room to motivation. For example, why is it important to have width bounds for fitting training data when the ultimate goal is generalization? Or if the NTK separation margin and minimum eigenvalue are the same for a finite sized network as they are in the infinite case, do we know that the finite sized network would generalize as well as the infinite? Maybe each result in the main body could be accompanied by a bit more of an explanation of why it’s fascinating or related to getting better performance out of neural networks.

**Questions:**

1. Are the citations you list around line 50 covering both why GD converges to arbitrarily small errors in nonconvex optimization and convergence results for  overparameterized networks? Are those two sentences supposed to be two disjoint sets of citations or are they related? Or was the first sentence just meant to introduce the related works and not be followed by citations?
2. It’s shown that ordinary relu networks can classify a dataset equivalently, but convex networks seem like they’re able to make discontinuous functions since neurons can ‘turn on’ to nonzero values. If the goal of the network is ultimately to generalize, might that matter that convex networks might learn discontinuous functions?

---

> ### Author Response · Authors · 2025-11-19
>
> We thank the reviewer for the valuable feedback and would like to respond as follows:
>
> 1-5) Thank you for the careful reading and pointing us to these typos which we will take care of in the next revision.
>
> 6) The contributions section/paragraph is meant as a roadmap and thus points to the contents of each section. The contribution (with the roadmap removed) are currently presented in only **two** bullets as:
>
> 1. We analyze theoretical properties of convex two-layer ReLU neural networks (NNs)
> 2. Using the established properties, we give an algorithm that approximates the parameter $\gamma$ and (we) improve the quadratic $m=\Omega(\gamma^{-2})$ width lower bound.
>
> Taking your and other reviewers comments into account, we think of rewriting this to disentangle the main contributions as follows:
> 1. We give a gradient descent algorithm that approximates the parameter $\gamma$.
> 2. We improve the quadratic $m=\Omega(\gamma^{-2})$ width lower bound.
>
> And a third contribution that comes 'along the way' is
>
> 3. We establish novel theoretical properties of convex two-layer ReLU neural networks (in particular in comparison to standard networks and in terms of the margin and eigenvalues of the NTK)
>
> and then we will finally list the more extensive roadmap comprising 6 items.
>
> 7) We could add a picture of alternately labeled points on a circle or on a 3-dimensional cube, if that is helpful. We could also add a table of notations somewhere (e.g. in the appendix).
>
> 8) Our main motivation is mentioned in a separate paragraph around lines 184-192. Yes, we can certainly add a bit more explanation or highlight why things are fascinating, given one additional page. However we do *not* agree that *our ultimate goal* is generalization. Studying the width of NNs has as its ultimate goal to establish matching upper and lower bounds on the required width to achieve zero gradient descent *training* error. That is a long standing line of research and is motivated by improving the efficiency of training rather than generalization.
>
> However, we do understand that the *reviewer's* ultimate goal might be generalization, and while most of the related literature does not walk down this path, we point to (Munteanu et al. 2022) as an example, whose main contribution are improved bounds on $\gamma$. Generalization bounds are only simple corollaries in that paper, because the generalization bounds they build upon (originally of Ji \& Telgarsky 2020) mainly rely on available bounds on $\gamma$ in the training setting. They can simply be replaced to give improved generalization as well. That highlights that the difficulty of improving (existent) generalization bounds indeed lies in showing improved *width* bounds. A similar observation was also mentioned in Remark 1.2 in (van den Brand et al. 2021). There it was mentioned any improved bound on the width directly improves their bound on training efficiency.
>
> Question 1. The first sentence was meant to introduce related work as a whole while the second sentence narrows the scope to the (extensively researched) over-parameterized setting.
>
> The two aspects are traditionally studied together, and the set of citations covers both aspects. The key is to choose the network width large enough to simulate (approximately) the convex infinite width case where zero training error becomes possible with gradient descent. But the width should of course stay small enough to keep memory and runtime low. Over-parameterization now concerns the case where the width has $m=poly(n)$ dependence, which is necessary in some settings studied in this paper.
>
> A few paragraphs later (around line 80) we also give a (less extensive) overview of the convergence of GD for networks of small width in the under-parameterized regime, where $m=polylog(n)$ is sufficient (but a $m\geq n$ dependence may still arise indirectly through the value of $\gamma$).
>
> Question 2. It is the *training* that has nicer properties (since the loss function is convex) than the original non-convex networks.
> The set of functions that can be represented by convex networks is equivalent as shown in Theorem 3.1 (first part). If the final non-convex network can represent some discontinuous function, then there exists a convex network that represents the same discontinuous function and thus also generalizes in the same way. The set of functions that convex networks can learn make up a superset, so switching to the convex case can only improve generalization if one trains to minimize some generalization error loss. It also depends on the data. Indeed we think that for real world data the target function the network is supposed to learn can be discontinuous.

---

> > ### Comment · Reviewer_cx59 · 2025-11-27
> >
> > Thank you to the authors for their careful responses to the above comments and for their revised PDF.  Although I do think the authors' changes have clarified several parts of their manuscript, there are still some reservations I have about the paper.  For instance, it requires almost 5 pages of introductory and background material, and then the remaining few pages of exposition are quite dense.  I also still share reviewer tse3's concern that the broader appeal of this work to the learning community may be limited.  I believe my original scores (and their descriptions) remain accurate - I do not have strong opposition but still struggle with these concerns.  I remain open to being persuaded either way during the reviewer-AC discussion period.

---

> > > ### Author Response · Authors · 2025-12-03
> > >
> > > We would like to follow up on the last comments of the reviewer as follows:
> > >
> > > - We understand that the density and long introductory part of the writeup requires some reorganization to make the main contributions and motivation even more clear and  upfront. We will take the advise seriously and improve the presentation to be more broadly accessible.
> > >
> > > - The comments and concerns of reviewer tse3 were against broader appeal due to the aforementioned presentation issues, **not** due to limited broader appeal of our work to the learning community! We believe that makes a crucial difference: as we have outlined in the related work, this line of research has started with an ICLR paper (Du et al., 2019a), main intermediate contributions were published in ICLR (Ji and Telgarsky, 2020) and ICML (Munteanu et al., 2022) papers (among several others). Following up on these, our work *closes* this line of research, and *motivates* adaptivity as a new branch of research, as it is the only hope to circumvent our lower bound. We thus believe that our work belongs equally well into the learning community.
> > >
> > > - Additionally, we have explained in our rebuttal what the connections to learning/generalization bounds are, and why they are rarely covered in this line of research (and in our paper).
> > >
> > > - If ultimately the reviewer has "no strong opposition", we believe that the reviewer's score should be adjusted to reflect a positive rather than negative bias.

---

### Official Review · Reviewer_tse3 · 2025-10-31

**Soundness:** 3
**Presentation:** 3
**Contribution:** 2
**Rating:** 4
**Confidence:** 2

**Summary:**

This paper proposes a new convex approximation for two-layer ReLU networks. The authors prove that their formulation is equivalent to the standard neural networks, thus allowing them to benefit from the common convex optimization analysis of the convergence. Overall, such a formulation is supposed to make the analysis of two-layer ReLU networks more intuitive and bridge some gaps in the existing worst/base case bounds.  Among the main results the authors provide is also a way to estimate the eigenvalue of the NTK in a sound way. Finally, several examples of tasks where the convex formulation is superior to the traditional one are shown.

**Strengths:**

1.	Novel study on the convex formulation of two-layer ReLU networks
2.	NTK perspective on the established equivalence and the derived proximity of the optimization dynamics.
3.	Examples of when the convex formulation can solve tasks that cannot be provably solved by the traditional two-layer ReLU network.

**Weaknesses:**

1.	Very dense paper that is hard to read
2.	Many key results are not compared to prior work
3.	The importance of certain contributions is not clear

**Questions:**

1.	Why did the authors choose a different name for the object of study? In the manuscript, they mention that they study gated ReLU networks, but call them convex two-layer ReLU networks.
2.	Could the authors provide a clear breakdown of their contributions when compared to prior works? While the authors cite relevant works (to the best of my knowledge), it is not immediately obvious to find a direct comparison to the results presented in them. Ideally, a reader would like to see a clear distinction after each major result, comparing it to existing results and/or explaining it novelty. Otherwise, the reader has to go through all cited papers to actually assess the importance of the authors’ findings.

---

> ### Author Response · Authors · 2025-11-19
>
> We thank the reviewer for the valuable feedback and would like to respond as follows:
>
> - We mention gated networks to make clear it is not our invention but there have been previous works introducing the same model before (although their focus/results were in different directions than ours). However, the main property and theme of *our* paper is *convexity*! As such the term 'convex NN' highlights the key property that make up or enable our results. On the other hand calling them 'gated NNs' is uninformative as it does not live up to any of the key properties that are important to our work.
>
> - We do not follow the comment that many key results are not compared to prior work. Some/most of our results are completely novel findings without predecessors in the literature, or we already explicitly state similar findings in related work and a comparison how our results differ or improve the previous results. We elaborate more below with a breakdown as requested (if indeed a mention is missing, we will add it to the manuscript)
>
> Section 3: The definition of cones is taken from Munteanu et al. 2022. The polynomial size upper bound $|S_0(X)|= O(n^{d-1})$ is known from classic literature (Cover 1965) which is already mentioned right before Lemma 3.2. The equivalence result (Theorem 3.1) is completely new, though similar convex formulations appeared in references by (Pilanci, Ergen, Mishkin) mentioned right before the Theorem! However, as discussed in the related work section their formulation is based on different duality approach and SDP cone programming rather than simple gradient descent as covered as a novel finding in our paper!
>
> Section 4: The finding that there exist convex networks that attain the same NTK margin and eigenvalues as their non-convex counterparts (Theorem 4.2) is completely new and important for Section 6.
>
> Section 5: The finding that gradients for the weight updates are equivalent is completely new and is important for Section 6 to allow gradient descent. Previous work mentioned above again only covered dual SDP cone programming not the simple GD!
>
> Section 6: The convergence result to $(1\pm\epsilon)$ approximate the NTK margin $\gamma$ is completely new. Evaluating the margin was not known before except for some specially constructed data (see strengths mentioned by Reviewer EEBd). Our results thus allow for simple numerical evaluation of the involved integrals with guaranteed accuracy.
>
> Section 7: For the same alternating points on the circle dataset a similar $\Omega(\gamma^{-2})$ lower bound had been proved in Munteanu et al. 2022. Our result is significantly stronger because it gives the same quadratic lower bound against *any* perfect NTK separator, not only against the standard used in upper bound analyses of Munteanu et al. 2022, Ji \& Telgarsky, 2020. This rules out any non-adaptive construction and indicates that novel adaptive techniques must be used to show (if it is possible) to get below quadratic. So our bound almost closes an important open problem, since adaptivity has never been explored in previous work (except for Lemma 3.5 resp. F.2 in (Munteanu et al. 2022) which is limited to the 2-dimensional case). However exploiting adaptivity in full generality is motivated as future direction.
>
> Finally the 3d Hypercube data had again been introduced in Munteanu 2022, but the expressivity distinction implied by it is completely new! In particular previous work on convex network formulations had not considered such distinction but only the other direction non-convex $\rightarrow$ convex.

---

> > ### Comment · Reviewer_tse3 · 2025-11-24
> > **Thank you for your clarifications**
> >
> > I would like to thank the authors for their clarifications. Could they be incorporated into the revised PDF and highlighted for better visibility?

---

> > > ### Comment · Reviewer_tse3 · 2025-11-26
> > >
> > > I went through the revised PDF, and it looks better to many now. I still feel that the paper is very hard to parse and may be very difficult to navigate for someone who discovers the reduction of regular two-layer NNs to their convex formulations for the first time (even the summary of contributions has many terms assuming very close familiarity with NTK terminology). I'll increase my score due to the authors' reply, but remain only partially convinced about the broader appeal of the presented contributions in their current shape.

---

> > > > ### Author Response · Authors · 2025-12-03
> > > >
> > > > We would like to thank the reviewer and promise that we will improve the overall accessibility to a broader audience.

---

### Official Review · Reviewer_trQH · 2025-11-01

**Soundness:** 2
**Presentation:** 1
**Contribution:** 2
**Rating:** 4
**Confidence:** 2

**Summary:**

This paper theoretically analyzes a convex variant of two-layer ReLU neural networksand establishes its close relationship to the standard, non-convex formulation. The key idea is to "convexify" the network by decoupling the neuron's activation pattern (determined by fixed "orientation" vectors) from its learned weight. This simplifies the optimization landscape, making the training problem convex, while retaining much of the expressive power of standard networks.

**Strengths:**

This paper theoretically analyzes a convex variant of two-layer ReLU neural networksand establishes its close relationship to the standard, non-convex formulation.

**Weaknesses:**

In my view, this paper could better isolate and clarify its key contributions to make them easier to grasp. Below are some of my concerns, which may be partially due to my unfamiliarity with the relevant literature.

The authors review certain aspects of convex two-layer neural networks and present some preliminary results. While these findings are somewhat interesting, they are not sufficiently compelling to convince me of the paper's overall strength.

Subsequently, the authors analyze the separation margin and the smallest eigenvalue of the NTK, claiming that when m=|S_{0}|, these quantities equal those of the convex neural network. I am unclear on the purpose or significance of this result.

Finally, they demonstrate that for some datasets, the required network width to approximate the NTK can be improved. If this is the major result, it should be stated explicitly and prominently at the beginning of the paper.

**Questions:**

same to the weakness.

---

> ### Author Response · Authors · 2025-11-19
>
> We thank the reviewer for the valuable feedback and would like to respond as follows:
>
> - We show that (under the usual assumptions of prior work) the two formulations are basically equivalent, in terms of their classifications (even more the functions they represent w.r.t. the data), gradient dynamics, NTK margin/eigenvalues etc. While these are a collection of single results that may not seem compelling, the entirety of these results act as a 'problem reduction' of training non-convex NNs to training convex NNs. Recall that if problem A reduces to problem B then an algorithm that solves B solves A as well. But here A seems more complicated to solve due to non-convexity and indeed this results in complicated convergence proofs with loss in parameters (in previous work). But now since we reduced to B which is convex, we do not even need to analyze any more, we can simply apply known GD analyses (Boyd, Nesterov, Bubeck) some of these GD results dating back to the sixties or earlier and hereby do not lose any parameter dependencies along the way. This finally means that the main source of difficulty lies solely in the initialization of the networks.
>
> - The purpose or significance is since the NTK margin and eigenvalue quantities are the same, we can now use a simple gradient descent to calculate them on convex networks to evaluate them for non-convex networks. Evaluating the margin was not known before except for some specially constructed data (see strengths mentioned by Reviewer EEBd). Our results thus allow for simple numerical evaluation of the integrals involved with guaranteed accuracy.
>
> - All three, 1) the aforementioned reduction as a whole (along the several smaller results), 2) the algorithmic result for calculating/approximating $\gamma$ within $(1+\epsilon)$, as well as 3) the new $m=\Omega(\gamma^{-2})$ lower bound are main results.
>
> - We note that $m=O(\gamma^{-2})$ is known unconditionally, $m=\Omega(\gamma^{-2})$ was known conditionally on taking an i.i.d. initialization w.r.t. the *Gaussian* measure.
>
> - Our new result is that $m=\Omega(\gamma^{-2})$ holds against *any* non-adaptive construction of an NTK separator. And we also state that this holds against *any analysis that we are aware of*, meaning that our new bound closes partially a long standing open problem.
>
> This is true, since adaptive construction has never been explored (as far as we know), but our result reveals that it is the only last hope for $m=O(1/\gamma)$ (or anything between linear and quadratic) upper bounds in the worst case setting. That is why we motivate such adaptive techniques as future directions in our conclusions.
>
> We will make this significance clearer and upfront by rewriting the outline and contributions of the paper.

---

> > ### Comment · Reviewer_trQH · 2025-11-21
> >
> > Thanks for your response. Though I am not fully convinced, I would like to increase my score accordingly.

---

> > > ### Author Response · Authors · 2025-12-03
> > >
> > > We would like to thank the reviewer and promise that we will incorporate these aspects into the manuscript.

---

### Official Review · Reviewer_EEBd · 2025-11-10

**Soundness:** 3
**Presentation:** 3
**Contribution:** 2
**Rating:** 6
**Confidence:** 3

**Summary:**

This work investigates a convex counterpart of a two-layer ReLU network and proves this convex variant shares similar properties with the original ReLU network such as the NTK matrices having the same eigenspectra as well as the same NTK margin. Additionally, the paper constructs a data distribution where any perfect NTK separator with width $O(\frac{1}{\gamma^2})$ must have weights dependent on initializion, improving the lower bound construction from prior work.

**Strengths:**

The construction of a dataset where a perfect NTK separator with subquadratic (with respect to the reciprocal of the margin) width must have weights dependent on initialization illuminates the difficulty of getting subquadratic width guarantees.

In addition, showing that GD on the convex variant converges arbitrarily close to the NTK margin is interesting as it provides an algorithm for computing such margins, something that was not known beyond special cases such as parity problems.

**Weaknesses:**

The convex variant of the ReLU networks seems quite tied to the 2-layer case and it does not seem easy to generalize this formulation to multi-layer ReLU networks, much less architectures with attention.

Furthermore, the convex formulations behave similarly to their original counterparts during training with GD only when the inner layer doesn't change much. In contrast, [1] analyzes GD on two-layer ReLU networks while also allowing for quite a bit of movement $O(\gamma \sqrt{m})$.

[1] Telgarsky, M. (2022). Feature selection with gradient descent on two-layer networks in low-rotation regimes. arXiv preprint arXiv:2208.02789.

**Questions:**

1. Are there any toy datasets where there exists linear width perfect NTK separator with initialization dependent weights but lack such a separator when the weights are initialization invariant?

2. Please provide comparison with [1] (specifically theorem 2.1 and 2.2). My main issue with the paper is the fact that the analysis of GD on the convex variant is only really useful when the inner layer doesn't move much. But [1] handles $O(\gamma \sqrt{m})$ amount of movement while also analyzing GD on the original formulation.

3. It seems the approximation algorithm for computing $\gamma_V$ (i.e. lemma 6.1) requires the knowledge of $\gamma_V$ as one needs to choose $B$ to be sufficiently large enough. Can you remove this circular dependency? I guess a doubling trick should work but that could be expensive if $gamma_V$ is tiny (e.g. if $\gamma_V$ is exponentially small with respect to dimension).


[1] Telgarsky, M. (2022). Feature selection with gradient descent on two-layer networks in low-rotation regimes. arXiv preprint arXiv:2208.02789.

---

> ### Author Response · Authors · 2025-11-19
>
> We thank the reviewer for the valuable feedback and would like to respond as follows:
> 1) We are not 100% sure we understand your question. Does initialization dependent/invariant weights refer to the weights that the perfect separator mapping $\bar v$ induces? If yes, we believe your question can be answered as follows:
>
> The alternating circular data set is the toy data you ask for, since our new Theorem 7.2 says that for this data, when you fix any perfect separator $\bar v$, then for any set of $m< \gamma^{-2}$ Gaussians, the induced weights $V$ will fail on at least one point with constant probability.
>
> On the other hand, Lemma 3.5 resp. F.2 in (Munteanu et al. 2022) constructs a weight matrix $V$ of size $m \approx \gamma^{-1}\log(n)$ adaptively to the Gaussian initialization to balance the number of vectors in each cone, so to get a perfect separator. This lemma holds for *any* $2$-dimensional data set, so it also holds for the alternating circular data.
>
> 2) While your reference [1] allows quite a bit of movement its analysis is mainly focused on parts that do not move significantly. More precisely [1] is using that the change of the inital weights is small and thus there always remains a good dircetion for the gradient. While this setting was not known to us before, it is interesting to cite it as a complementary direction.
>
> Our main goal is to reduce to a variant of neural networks that simplifies or circumvents the convergence analysis and helps localizing or isolating the difficulty to the initialization step for proving better lower bounds.
>
> Overall one could say these are two complementary approaches to analyze two layer ReLU networks.
>
> 3) Indeed we cannot remove the circular dependency. However, trying multiple values for $\gamma$ does not increase the running time by a large amount as the training of small networks is faster and requires less iterations than networks with larger width. There might also be heuristics that could help in practice. For example if we fail for some value $m$, we could reuse the already trained network and just add more neurons to the middle layer. This way we could reuse the progress made even if $m$ was chosen too small.

---

> > ### Author Response · Authors · 2025-12-03
> >
> > We would like to clarify and follow up on our previous comment on question 2:
> > - As we had written, the analysis given in reference [1], even though it allows more movement that standard NTK analyses, still focuses on parts that have bounded movements. We want to clarify that this can be used totally analogously to the standard NTK assumptions to derive similar equivalence results.
> >
> > - The reviewer had asked for a direct comparison of theorem 2.1 and 2.2 in [1], and we had only written that they are complementary. Indeed the two theorems are **upper** bounds of $1/\gamma^2$ for SGD and gradient flow. In that respect, they are comparable to the earlier GD and SGD $1/\gamma^2$ **upper** bounds of (Munteanu et al. 2022). We have thus added [1] to the corresponding part (around line 90) of our related work section. *Our* new result (Theorem 7.2) is a matching $1/\gamma^2$ **lower** bound, corroborating the tightness of both these previous works, (in fact it holds against all previous works known to us), and is in that regard complementary to both, see also the added discussion below Theorem 7.2 (lines 504-510).

---

### Author Response · Authors · 2025-11-26
**Submission updated**

We would like to thank all reviewers again for their valuable comments, and for reconsidering their evaluation based on our response.

As suggested by Reviewer tse3, we have meanwhile updated our submission according to the discussion so far. We hope that the revision clarifies or resolves all main concerns. All changes are highlighted in red in the manuscript.

---

### Author Response · Authors · 2025-12-03
**Final comments**

Dear AC,

it is a bit unfortunate that the raised scores we received in response to our rebuttal and PDF revision have been set back to their initial state. It is also unfortunate that we could not continue to discuss with the same reviewers, since it seems a few aspects remained open.

However, we hope that you will be able to follow the previous discussions. Please let us know should there be any remaining questions we could help to clarify.

We would like to add a few clarifications and responses to the latest developments, that we kindly ask you to take into account towards your final assessment.

Thank you, the authors.

---

### Meta-Review · Area_Chair_eANX · 2026-01-06

**Summary:**

This paper analyzes the classification error of a convex two-layer ReLU network under a separability assumption, using fixed first-layer features and optimizing only the output weights. As a result, the model effectively reduces to a linear classifier over a fixed feature map. Consequently, the theoretical analysis largely follows standard arguments for linear models, which substantially limits the technical novelty and significance of the results.

**Reviewer Concerns:**

Several reviewers raised concerns regarding the organization and clarity of the paper, as well as the focus of the technical exposition. In particular, substantial space is devoted to results that are either trivial or not essential to the main claims (e.g., explicitly proving convexity of the loss despite the model being linear). Moreover, since the analysis ultimately reduces to linear-model arguments, a number of derivations and equations appear unnecessary for establishing the stated results. These issues contribute to a lack of clarity in identifying the paper’s core technical contribution.

**Reviewer Scores:**

Addressing the above concerns would require at least a major revision, including a clearer articulation of technical novelty beyond linear models and a significant reorganization of the presentation. Given this, most reviewers are unlikely to revise their scores upward in the current form.

---

### Decision · Program_Chairs · 2026-01-26

Reject